# Physical control of the northern Arabian Sea winter chlorophyll bloom interannual variations

5 M.G. Keerthi[1], Matthieu Lengaigne[2, 3], Marina Levy[2], Jerome Vialard[2], V. Parvathi[1], Clément de Boyer Montégut[4], Christian Ethé[2], Olivier Aumont[2], I. Suresh[1], V.P. Akhil[1], P.M. Muraleedharan[1]

[1] CSIR-National Institute of Oceanography (CSIR-NIO), Goa, India

[2] Sorbonne Universités (UPMC, Univ Paris 06)-CNRS-IRD-MNHN, LOCEAN Laboratory, IPSL, Paris, France

[3] Indo-French Cell for Water Sciences, IISc-NIO-IITM–IRD Joint International Laboratory, NIO, Goa, India

[4] IFREMER, Univ. Brest, CNRS, IRD, Laboratoire d'Océanographie Physique et Spatiale, IUEM, 29280, Brest, France

*Correspondence to*: M. G. Keerthi (keerthanaamg@gmail.com)

**Abstract.** The northern Arabian Sea hosts a winter chlorophyll bloom, triggered by convective overturning in response to
15 cold and dry northeasterly monsoon winds. Previous studies of interannual variations of this bloom only relied on a couple of
years of data and reached no consensus on the associated processes. The current study aims at identifying these processes
using both ~10 years of observations (including remotely-sensed chlorophyll data and physical parameters derived from
Argo data) and a 20-year long coupled biophysical ocean model simulation. Despite discrepancies in the estimated bloom
amplitude, the six different remotely-sensed chlorophyll products analysed in this study display a good phase agreement at
20 seasonal and interannual timescales. The model and observations both indicate that the winter bloom interannual fluctuations
are strongly tied to mixed layer depth interannual anomalies (~0.6 to 0.7 correlation), which are themselves controlled by the
net heat flux at the air-sea interface. Our modelling results suggest that the mixed layer depth control of the bloom amplitude
ensues from the modulation of nutrient entrainment into the euphotic layer. In contrast, the model and observations both
display insignificant correlations between the bloom amplitude and thermocline depth, which precludes a control of the
25 bloom amplitude by daily dilution down to the thermocline depth, as suggested in a previous study.

# 1. Introduction

Located in the western arm of the northern Indian Ocean, the Arabian Sea (AS) is forced by energetic seasonally reversing monsoon winds, which largely control its physical properties. During boreal summer, strong southwesterly winds blow over the western AS (Findlater, 1969) and cause intense upwelling along the coasts of Somalia and Oman and downwelling in the central AS (e.g. Schott and McCreary, 2001). During boreal winter, the Eurasian continent cools and a high-pressure region develops on the Tibetan plateau, resulting in cold and dry north/northeasterly winds (e.g. Smith and Madhupratap, 2005) and leading to strong evaporative cooling (Dickey et al., 1998). These diverse physical processes cause substantial variations in marine biogeochemical and ecosystem response. Being one of the most productive regions in the world ocean (Satya Prakash and Ramesh, 2007; Prasanna Kumar et al., 2000) and being home to the second most intense oxygen minimum zone in the world ocean (Kamykowski and Zentara, 1990), the AS provides an excellent test bed for studying biophysical coupled processes (McCreary et al., 2009).

Previous studies have extensively described the seasonal variability of surface chlorophyll (hereafter, SChl) in the AS. The AS biogeochemical properties vary from stratified oligotrophic conditions during inter-monsoon periods to eutrophic conditions during monsoons (Smith et al., 1998; McCreary et al., 2009). Neither surface irradiance nor temperature limits the biological productivity in this tropical basin: instead, it is mostly attributed to dynamical processes in response to the monsoonal forcing (e.g. Barber et al., 2001; Marra and Barber, 2005). During boreal summer, the largest seasonal blooms are found along the coasts of the Arabian peninsula (e.g. Banzon et al., 2004; Lévy et al., 2007; Wiggert et al., 2005) and are exported offshore by mesoscale eddy stirring (e.g. Resplandy et al., 2011). In boreal winter, convective overturning allows entrainment of nutrients into the mixed layer and leads to a prominent bloom in the northern AS (Banse and English, 2000; Madhupratab et al., 1996; Prasanna Kumar et al., 2001; Wiggert et al., 2002). In addition to these seasonal variations, several studies revealed large interannual variations in the AS winter chlorophyll from either satellite (Banse and McClain, 1986; Banse and English, 1993; Sarma et al., 2006; Wiggert et al., 2002; Sarma et al., 2012) or in-situ measurements (Bauer et al., 1991; Madhupratap et al., 1996; Gundersen et al., 1998; Prasanna Kumar et al., 2001). This strong interannual variability of the northern AS winter bloom is illustrated on Fig.1a,b for two consecutive winters. A particularly intense bloom developed in the northern AS during winter 2007 (Fig. 1b), with high SChl concentration (>1.0 mg.m$^{-3}$) extending southward down to 14$^{o}$N. In contrast, the winter 2006 bloom remained confined to the northern AS (Fig. 1a), with high chlorophyll concentration (>1.0 mg.m$^{-3}$) limited to the north of 20$^{o}$N. The difference in the amplitude of the bloom between winter 2006 and 2007 averaged over the northern AS box (hereafter NAS region shown in Fig. 1) reaches 0.22 mg.m$^{-3}$, which is approximately 30% of the climatological winter chlorophyll value.

Understanding the mechanisms driving these chlorophyll interannual variations is important, as this might have a profound influence on the variations of the fish stocks and of the oxygen minimum zone in the AS. To date, only a few

studies have discussed the mechanisms that could be responsible for the winter bloom interannual fluctuations (Banse and McClain, 1986; Prasanna Kumar et al., 2001; Wiggert et al., 2002) and no consensus has been reached so far. Comparing in-situ time series in February 1995 and 1997, Prasanna Kumar et al. (2001) suggested that increased convective cooling resulted in an intense convective mixing, a deeper mixed layer depth (hereafter, MLD), enhanced nutrients injection through entrainment and ultimately a stronger bloom in winter 1997 than in winter 1995. Such mechanism hence implies that the interannual MLD variability should be positively correlated with the interannual SChl variability (the so-called "Bermuda paradigm"). Keerthi et al. (2016) found large winter MLD variations in the NAS over the past two decades, largely driven by fluctuations in the advection of dry, cold air from the continent, but did not investigate their biogeochemical consequences. Comparing three consecutive winters from 1998 to 2000, Wiggert et al. (2002) suggested using a simple one-dimensional model, that interannual variations of the bloom intensity were controlled by the night-time penetration of diurnal mixing, whose maximum downward penetration is constrained by the thermocline depth (hereafter, TCD). In this paradigm, a deeper TCD allows for a deeper night-time mixing, a greater dilution of phytoplankton biomass and stronger inhibition of the bloom development. This alternative scenario hence implies that the interannual TCD variability should be negatively correlated with the interannual SChl variability in the northern AS during winter, a relationship that directly contradicts the Bermuda paradigm as pointed out by Wiggert et al. (2005). Prasanna Kumar et al. (2001) and Wiggert et al. (2002) hence proposed two conflicting mechanisms, which respectively imply a positive correlation between MLD and SChl interannual variations and a negative correlation between TCD and SChl interannual variations. These papers however based their conclusions on the analysis of rather small samples, i.e. two one-month-long in-situ time series for Prasanna Kumar et al. (2001) and three consecutive winters from satellite chlorophyll data for Wiggert et al. (2002). The absence of consensus on the processes responsible for the NAS winter bloom interannual variations hence pleads for additional studies, especially now that longer remotely-sensed and in-situ time series are available.

In the present paper, we hence aim at better assessing and understanding the interannual variability of the NAS winter bloom. On the observational side, this study benefits from the extended temporal coverage of the satellite chlorophyll data (~15 years) and the advent of the Argo program that allows monitoring in-situ MLD and TCD variations from 2002 onwards. Performing a combined analysis of these datasets allowed us to perform a direct comparison between these physical parameters and the chlorophyll variability. In addition to these satellite and in-situ observations, we analyse outputs from a ~20 year long coupled biophysical model simulation. The analysis of this simulation, which accurately simulates NAS winter chlorophyll interannual variations, allows us to investigate the subsurface processes not readily available from observations. Section 2 describes the observational products (satellite chlorophyll estimates and Argo-derived MLD and TCD; Sect. 2.1) and the numerical experiment (Sect. 2.2). Section 3 provides an intercomparison of the available satellite chlorophyll products over the NAS (Sect. 3.1) and the model evaluation (Sect. 3.2). Section 4 provides a description of the chlorophyll interannual variability, its relationship with physical parameters and discusses the mechanisms driving these fluctuations. Section 5 finally provides a summary and discussion of our results.

## 2.    Data and Method

### 2.1.  Observations

The SChl estimates analysed in the present study are derived from different instruments (SeaWiFS, MERIS and MODIS), retrieval algorithms and span different periods (Table 1). We will compare these different retrievals in Sect. 3.1, in order to assess the robustness of remotely-sensed data to investigate the NAS winter bloom. We used the Level-3 Standard Mapped Images with a 9x9-km spatial and a monthly temporal resolution downloaded from http://oceancolor.gsfc.nasa.gov for all of these single-mission products.  In addition, we also used three level 3 merged ocean-color products downloaded from http://www.oceancolour.org/ at 4x4-km and monthly resolution: the weighted average empirical (AVW) product, the semi-analytical Garver Siegel Maritorena (GSM) product and the Ocean Color Climate Change Initiative (OC-CCI) product. The longest observational period is provided by the OC-CCI product and spans from October 1997 to December 2013.

The ocean physical parameters are derived from an updated version of the dataset described in Keerthi et al. (2013), with an extended temporal coverage (2002 to 2013) and an estimate of the TCD in addition to that of the MLD. These MLD and TCD datasets are built from a combination of Argo and historical temperature and salinity profiles. To assess whether the mechanism proposed by Prasanna Kumar et al. (2001; i.e. the Bermuda paradigm) dominates the interannual variability of the winter bloom in the northern AS over this extended period, we will investigate if there is a correlation between in-situ-derived interannual MLD and the satellite-derived interannual SChl anomalies. We will test the alternative mechanism proposed by Wiggert et al. (2002) by investigating if there is a negative correlation between interannual in-situ-derived TCD anomalies and interannual satellite-derived SChl anomalies in the northern AS during winter. MLDs were estimated using a temperature criterion, and are defined as the depth where the temperature increases by 0.2°C with respect to the temperature at 10 m. The reference depth was taken at 10 m to avoid aliasing by the diurnal cycle. The TCD was defined as the depth of the maximal vertical temperature gradient. MLDs and TCDs were estimated from individual temperature profiles at their native vertical resolution. The resolution of the data was then degraded to a regular $2^o$ monthly grid, by taking the median of all MLDs and TCDs in each grid mesh. A more detailed description of this procedure can be found in Keerthi et al. (2013). An overview of the spatio-temporal coverage of this dataset over the NAS is provided in Fig. 2. While the data coverage is particularly sparse in winter before 2002 in our targeted region (e.g. less than 10 data per month are available in winter 2000 in the NAS region), the data density increased considerably after 2002, with the development of the Argo program (Fig. 2a). After 2002, the NAS box winter data density ranged from 25 profiles per month during 2005 to nearly 120 profiles per month during 2012. This implies that the interannual MLD/TCD values averaged over the NAS box during winter 2002 to 2013 are built from an average of 100 to 500 individual values, giving us confidence in the robustness of the interannual MLD/TCD variability derived from this in-situ dataset. It should be noticed that the data coverage is however not spatially

homogeneous, with the highest coverage along a shipping line crossing the NAS box (Fig. 2b).

We also use the World Ocean Atlas (WOA13) climatology (Boyer et al., 2013) to derive climatologies of the thermocline and nitracline depths, calculated as the depths of maximum temperature and nitrate gradients, respectively. Wind speeds, surface air temperatures and net heat fluxes are derived from the Tropflux product (Praveen Kumar et al., 2012). In order to assess the variability associated with various interannual climate modes, we have used standard climate indices. El Niño-Southern Oscillation (ENSO) is represented using the Nino3.4 index, which is the averaged SST anomalies over the Niño3.4 (120–170°W, 5°N–5°S) region during November-January, available from http://www.cpc.ncep.noaa.gov /products/analysis_monitoring/ensostuff/detrend.nino34.ascii.txt. The Indian Ocean Dipole (IOD) is represented by the dipole mode index (DMI, Saji et al., 1999), computed as the difference between interannual SST anomalies in the western (50–70°E, 10°N– 10°S) and eastern (90–110°E, 10–0°S) equatorial Indian Ocean during September–November, available from http://www.jamstec.go.jp/frcgc/ research/d1/iod/DATA/dmi.monthly.txt.

## 2.2. Model configuration and numerical experiments

These observational products are complemented by a biophysical model simulation, which allows extending our analysis over a longer time period and analysing depth-integrated biogeochemical properties that are not captured by satellites. We use the NEMO (Nucleus for European Modelling of the Ocean; see Madec (2008) for an exhaustive description) ocean general circulation model coupled with the latest version of PISCES (Pelagic Interaction Scheme for Carbon and Ecosystem Studies; see Aumont et al. (2015) for an exhaustive description of the model) biogeochemical component. Briefly, PISCES includes two sizes of sinking particles and four "living" biological pools, that represent two phytoplankton (nano-phytoplankton and diatoms) and two zooplankton (micro-zooplankton and meso-zooplankton) size classes. Phytoplankton growth is limited by five nutrients: $NO_3$, $NH_4$, $PO_4$, $SiO_4$, and Fe. The ratios among C, N, and P are kept constant for the "living" compartments, at values proposed by Takahashi et al. (1985). On the other hand, the iron, silicon and calcite pools of the particles are explicitly modelled. As a consequence, their ratios are allowed to vary. Nutrients are supplied to the ocean from five different sources: atmospheric dust deposition, rivers, sea-ice, sediment mobilization, and hydrothermal vents. An interannually varying dust deposition dataset is not available to date. Dust deposition from the atmosphere is hence estimated from climatological monthly deposition maps simulated by the National Centre for Atmospheric Research model (Mahowald et al., 2005), assuming constant values for the iron content and solubility (Moore et al., 2004). This choice is further justified by the modelling results of Aumont et al. (2008) who demonstrated that the variability of SChl induced by the interannual variability of aerial iron deposition is likely to be very small everywhere especially relative to the impact of the ocean dynamics, because largest fluctuations of surface iron produced by dust occur in oligotrophic regions where phytoplankton growth is not primarily controlled by iron availability. The internal Fe contents of both phytoplankton groups and Si contents of diatoms are prognostically simulated as a function of ambient

concentrations in nutrients and light level. Details on the red-green-blue model by which light penetration profiles are calculated, are given in Lengaigne et al. (2007). The Chl/C ratio is modelled using a modified version of the photo-adaptation model by Geider et al. (1998). For a more detailed description, manuals for NEMO and PISCES are available online at http: //www.nemo-ocean.eu/About-NEMO/Reference-manuals.

The regional configuration used in this study is an Indian Ocean sub-domain of the global ¼° resolution (i.e. cell size ~25 km) configuration described by Barnier et al. (2006). It has 46 vertical levels, with a resolution ranging from 5 m at the surface to 250 m at the bottom. The African continent closes the western boundary of the domain. The oceanic portions of the eastern, northern and southern boundaries use radiative open boundaries (Treguier et al., 2001), constrained with a

10 150-day relaxation timescale to outputs from a global simulation (Dussin et al., 2009). The circulation and thermodynamics of this regional configuration have been extensively evaluated and reproduce observed variations of key physical parameters well in several Indian Ocean regions (Vialard et al., 2013; Akhil et al., 2014, 2016; Praveen Kumar et al., 2014), including the AS (Nisha et al., 2013; Keerthi et al., 2016).

The simulation starts from rest, with temperature and salinity initialized from the WOA13. PISCES biogeochemical tracers are also initialized from the WOA13 database for nutrient and from the climatology of a global simulation for the other tracers (Aumont and Bopp, 2006). After 5 years of spin-up with a climatological forcing, the model is forced with the Drakkar Forcing Set #4.4 (DFS4.4, Brodeau et al., 2010) from 1980 to 2012. This forcing is a modified version of the CORE dataset (Large and Yeager, 2004), with atmospheric parameters derived from ERA40 reanalysis (Uppala et al., 2005) and

ECMWF analysis after 2002 for latent and sensible heat fluxes computation. Radiative fluxes are taken from the corrected International Satellite Cloud Climatology Project-Flux Dataset (ISCCP-FD) surface radiations (Zhang et al., 2004) while precipitation forcing are a blend of satellite products described in Large and Yeager (2004). All atmospheric fields are corrected to avoid temporal discontinuities and to remove known biases (see Brodeau et al., 2010 for details). In the following, we will analyse the 1993-2012 period.

### 3. Evaluation of the interannual variability in the Northern Arabian Sea

In this section, we provide an intercomparison of the six ocean colour products described above (Sect. 3.1) and a brief description on model performance at seasonal and interannual time scales in our targeted region (Sect. 3.2).

### 3.1. Satellite SChl products intercomparison

One of the major limitations of ocean colour imagery is the inability to perform accurate retrievals under clouds and in presence of aerosols. In the AS, this is particularly challenging during the summer monsoon but not during winter when

the data coverage is larger than 90% for all datasets considered, and reaches 98% in the case of the merged products (Table 1). Fig. 3a shows the SChl climatological seasonal cycle averaged over the NAS region for the six available satellite products. This figure reveals a good agreement between the different products in terms of seasonal phasing, with a semi-annual cycle associated with two seasonal blooms, one in summer (maximum in July, except for MERIS) and the other in winter (maximum in February for all products). The amplitudes of these seasonal blooms clearly differ depending on the product, with largest blooms in MODIS (up to 2 mg.m$^{-3}$ in winter and 5 mg.m$^{-3}$ in summer) while seasonal blooms hardly reach 1 mg.m$^{-3}$ in GSM, OC-CCI and MERIS. However, the amplitude of the summer chlorophyll bloom is uncertain, given the less data coverage during summer, especially in July where there is least data coverage.

Fig. 3b allows assessing the amplitude of the winter bloom interannual variations in each product by computing the standard deviation of the interannual chlorophyll variations for each calendar month from October to May. This figure shows that the largest interannual deviations from the climatological evolution depicted on Fig. 3a occur during February and March, while a minimum in the amplitude of interannual chlorophyll variability is generally found during November and April. Based on this seasonality, we define winter in the following as the period encompassing the large climatological and interannual SChl signals, i.e. from December to March.  For instance, we will refer to winter 2002 as the period averaged from December 2002 to March 2003. Fig. 3b however also illustrates that the amplitude of the winter bloom interannual variations considerably varies amongst SChl products. MODIS displays the largest winter standard deviation (up to 0.6 mg.m$^{-3}$ in March) and SeaWiFS the weakest one (up to 0.2 mg.m$^{-3}$ in March). This indicates that the analysis of interannual SChl fluctuations may heavily depend on the product considered.

Scatterplots of monthly SChl interannual anomalies from the different products are shown on Fig. 4. Despite varying amplitudes amongst products, there is generally a good phase agreement between the monthly anomalies from the different products, with correlation ranging from 0.55 between AVW and MODIS to 0.98 between OC-CCI and MERIS. Amongst the three merged products, OC-CCI displays the best match with the three individual satellite products, with a correlation of 0.84, 0.98 and 0.91 with MODIS, MERIS and SeaWiFS respectively. The amplitude of anomalies from the merged products generally matches that of MERIS (with regression coefficients ranging from 0.87 to 1.44) but are considerably lower than the ones estimated by MODIS (regression coefficients ranging between 0.23 and 0.42). In the following, most results will be illustrated with the OC-CCI product that displays the best phase agreement with the three individual satellite products (Fig. 4), offers a very good coverage (Table 1) and spans the longest period (1997-2013). It must however be kept in mind that the amplitude of the interannual SChl anomalies remains uncertain, given the large discrepancies amongst products. In the following, we however show that the interannual relationships existing between SChl and ocean physical parameters are generally robust amongst ocean color products.

### 3.2. Model evaluation

A brief evaluation of the seasonal cycle of SChl in the model simulation follows. The model accurately captures the large-scale SChl patterns in the AS for both summer (Fig. 5a, c) and winter (Fig. 5b, d). As for observations, the largest SChl bloom occurs in summer along the Oman coast while the winter bloom is maximum in the northernmost part of the AS (Fig. 5b, d). The modelled seasonal SChl evolution agrees well with the OC-CCI data in terms of amplitude and timing (Fig. 6a, d), with a clear semi-annual cycle characterized by a larger SChl bloom during summer (up to 1.5 mg.m$^{-3}$) than in winter (up to 1 mg.m$^{-3}$) and minimum SChl concentrations (less than 0.5 mg.m$^{-3}$) during inter-monsoons. Even though the model reasonably simulates the amplitude and timing of the SChl bloom, the summer bloom maximum in the NAS occurs two months later in the model (September) as compared to observations (July). The seasonal timing of the winter bloom in the NAS is however very accurately captured, with a maximum bloom occurring in February in both model and observations (Fig. 6a, d). This winter bloom occurs in response to convective vertical mixing and related MLD deepening (Fig. 6a, b) driven by the cold, dry northeasterly winds (McCreary and Kundu, 1989; Madhupratab et al., 1996; Prasanna Kumar et al., 2001; Wiggert et al., 2000; Lévy et al., 2007; Kone et al., 2009). The model captures the main features of subsurface physical and biogeochemical variations (Fig. 6b, e): the winter bloom is triggered by the deepening of the mixed layer associated with a strong cooling by surface heat fluxes (Fig. 6c, f), accompanied by a deepening of the thermocline and nitracline. The maximum MLD deepening occurs in January in the model and observations (Fig. 6b, e), one month before SChl peak (Fig. 6a, d). From then on, the upper ocean restratifies and the MLD shoals in response to increased net heat fluxes into the ocean (Fig. 6c). This MLD shoaling combined with a nitracline that remains deep (Fig. 6b) limits further nitrate supply to the MLD. This analysis briefly illustrates the ability of the model to capture the main biogeochemical features and related mechanisms in the NAS in winter.

This simulation not only reasonably captures the SChl seasonal cycle in the NAS region but also its winter interannual variability. Fig. 1 provides a first illustration of the model's ability to capture the amplitude of the contrasted surface blooms during the 2006 and 2007 winters discussed in the introduction. In agreement with observations, the simulation displays a winter bloom that extends further south in 2007 than in 2006, resulting in larger mean SChl concentrations over the NAS region in 2007. Fig. 7a provides a more thorough validation of the model interannual SChl variations in this region. Observed winter SChl interannual anomalies range from +0.4 mg.m$^{-3}$ (winter 2011) to -0.3 mg.m$^{-3}$ (winter 2012; Fig. 7a). The largest observed winter positive anomalies are found during 2011, 2007, 2002, 2000 and 1999 (Fig. 7a), while the strongest negative anomalies are found in 1997, 2003, 2006, 2008, 2010 and 2012 (Fig. 7a). The modelled NAS winter interannual SChl anomalies agree generally well with those from the OC-CCI dataset both in terms of phase and amplitude (Fig. 7a), with a correlation between the two time series reaching 0.69 over the 2001-2011 period and 0.52 over the 1997-2011 period, both significant at the 99% confidence level. The main mismatch is found during 1997 and 2002 when the model and observational datasets display opposite anomalies. The observed MLD also exhibits large

fluctuations, ranging from -23m in winter 2006 to around ~+14m in winter 2001, 2007 and 2011 (Fig. 7b). The model is also able to capture these observed MLD interannual variations (Fig. 7b), with a 0.65 correlation over the 2001-2011 period, significant at the 95% confidence level. The main disagreement between the model and observations occurs during the winters of 2002 and 2008, where the observed signals are not well captured by the model. Finally, the observed TCD also exhibits large year-to-year variations in winter, ranging from -15m in winter 2007 and 2009 to ~+15m in winter 2001. In contrast to SChl and MLD, the model does not capture the observed TCD variability well (0.3 correlation), although some major events such as the thermocline shoaling in 2007 and 2009 and the deepening in 2011 are properly simulated. However, the good agreement between the modelled and observed interannual SChl variability in NAS allows us to confidently use the model over a longer period (1993-2012) to further investigate interannual chlorophyll variability and its driving mechanisms.

### 4. Physical drivers of the interannual SChl variability

In this section, we describe how the main characteristics of the interannual chlorophyll variations in the NAS relate to ocean physical properties (MLD, TCD). The hypotheses of Wiggert et al. (2002) and Prasanna Kumar et al. (2001) for the mechanisms that controls SChl interannual variations imply a correlation of SChl anomalies with TCD and MLD anomalies, respectively. In order to test those hypotheses, we compared the time evolution of the OC-CCI SChl, MLD and TCD anomalies in the NAS box from 2002 to 2013 in Fig. 8. This figure illustrates that observed interannual SChl anomalies are closely related to interannual MLD fluctuations, with deeper MLDs generally associated with a positive chlorophyll anomaly, and vice-versa. This is verified for most winters, except for 2002 and 2004, where positive chlorophyll anomalies are concomitant with a modest shoaling. In addition, there is a consistent time lag between the MLD and SChl anomalies, with MLD anomalies usually peaking in February and chlorophyll anomalies peaking one month later. In contrast, there is no obvious connection between TCD and SChl anomalies (Fig. 8b): positive SChl anomalies can be either associated with thermocline deepening such as in 2011 or a thermocline shoaling as in 2002 and 2007. Similarly, negative SChl anomalies can be associated to a thermocline deepening as in 2006 or to a shoaling as in 2008 and 2012.

A more quantitative examination of the relationship between interannual SChl anomalies and MLD/TCD anomalies is provided in Fig. 9. As shown in Fig. 9a,c, the winter-averaged SChl and MLD anomalies are strongly correlated in both observations (0.72, statistically significant at the 99% confidence level; Fig. 9a) and the model (0.59 correlation significant at the 99% confidence level; Fig. 9c). In contrast, there is no statistically significant correlation at the 90% confidence level between SChl and TCD variations over the same period for the two datasets (Fig. 9b,d). The dependency of these relationships to the ocean color product is shown in Table 2. This table indicates that all observational products display larger correlations between SChl and MLD than between SChl and TCD anomalies. The strength of the MLD/SChl relationship however varies depending on the product considered, with the largest correlation for MODIS (0.86) and the weakest for the AVW product (0.38, not significant at the 90% significance level). None of the SChl products exhibits a

significant relationship with TCD at the 90% confidence level (Table 2). These results are a strong indication that interannual SChl variations are controlled by MLD rather than by TCD variability.

The spatial distribution of the typical SChl, MLD and TCD anomalies during an anomalously strong bloom event is shown in Fig. 10 for both observations and model. This composite pattern is constructed from the half-difference between positive and negative events highlighted on Fig. 7 for model and Fig. 8 for observations. Observations and model exhibit very similar spatial patterns: the maximum SChl anomaly signal (exceeding 0.5 mg.m$^{-3}$) occupies the northern part of the box around 21°N, 64°E (Fig. 10a,e), with weaker but still significant SChl signals found everywhere in the NAS box. This SChl pattern matches well with the MLD pattern, with maximum MLD positive anomalies (exceeding 16 m) occurring at the northern boundary of the NAS box (Fig. 10b,f) and significant positive MLD anomalies everywhere in the NAS box. In contrast, the TCD composite hardly shows any significant anomaly within the NAS box during an anomalous strong bloom (Fig. 10c, g). This composite analysis hence confirms that the relation between SChl and MLD interannual variations (and absence of relation with TCD) deduced from Figure 8 NAS-averaged values holds over the entire region.

The availability of chlorophyll and nitrate data at depth from the model allows going a step further in the description of the processes driving the chlorophyll variability. Fig.11a-d shows the chlorophyll and nitrate evolution between 100 m and the surface, averaged over the NAS box for the two contrasted winters of 2006 and 2007, already discussed on Fig. 1. Both years exhibit a chlorophyll bloom in winter, with maximum chlorophyll concentration in the surface layers in February. The absence of winter deep chlorophyll maximum (DCM) precludes the entrainment of chlorophyll from below to be responsible for the SChl bloom during this season. This is clearly visible from Fig. 11a,b, which shows that the increase in chlorophyll at the surface layer is not associated with a vertical redistribution of chlorophyll, but with an increase in the vertically integrated biomass. Let us now investigate what caused this larger phytoplankton concentration during winter 2007. The MLD is deepest in January and reaches ~ 50 m during both winters. The similar maximum winter MLDs during the two years induce a similar supply of subsurface nutrients (the nitrate concentration 10 m below the MLD, a proxy for the nitrate content of the water entrained or mixed into the mixed layer, is very similar for both years until January: Fig. 11gh). This yields a very similar nitrate concentration in January 2007 (7.23 mmol.m$^{-3}$) and January 2008 (7.29 mmol.m$^{-3}$). Consequently, SChl concentrations are very close in January (Fig. 11ef). The main difference between the two winter blooms is their duration: the 2006-07 bloom was over in March, while the 2007-08 bloom still persisted (Fig. 11). This is associated with a MLD that remained deep until February (~50m) in 2008, whereas it started shoaling one month earlier in 2007. The bottom of the deeper MLD in February 2008 is closer to the nutrient-rich subsurface layer, sustaining a larger nutrient input through turbulent fluxes. As a result, the mixed layer nitrate concentration reaches ~8 mmol.m$^{-3}$ in February 2008 against ~4 mmol.m$^{-3}$ in 2007 (Fig 11g,h). In February of both years, those nitrate concentrations are high enough so that phytoplankton growth is not nutrient-limited, which explains the similar SChl concentrations (~1.2 mg.m$^{-3}$) during that month (Fig 11e,f). Nitrate becomes limiting in March during both years, yielding no further biomass production after March.

It however takes more time for phytoplankton to exhaust the larger February 2008 mixed layer nitrate content, allowing for the 2008 bloom to persist until March. It must finally also be noticed that differences in nitracline depths cannot explain the bloom differences between these two winters: the nitracline is indeed slightly deeper in 2007-08 than in 2006-07 (red lines on Fig. 11 a-d). Overall, the comparison of these two years supports the important role of the February-March MLD variations in setting the near-surface nutrient content and chlorophyll value.

Fig. 12 allows exploring if the processes observed for the 2006 and 2007 contrasted winters also operate to explain SChl winter anomalies over the entire period. As the largest winter MLD and SChl variability occurs in February-March (see Fig. 7 and 8) the analysis in Fig. 12 is restricted to the February-March period. Fig. 12a shows that the 0-200 m integrated chlorophyll anomalies exhibit an even stronger relationship with MLD fluctuations (0.84 correlation) than with SChl (Fig. 9c; 0.6 correlation), demonstrating that SChl variability does not arise from a vertical redistribution of chlorophyll within the water column, but mainly results from phytoplankton growth. In addition, larger MLD interannual anomalies are associated with more nutrients in the mixed layer over the 20-year period analysed (Fig. 12b), with a 0.63 correlation between the two parameters. This can occur either through a modulation of the maximum MLD and hence of the amount of nutrients entrained into the mixed layer, or through the period when the MLD is deep (as for 2006/2007) and hence through turbulent fluxes of nutrients into the MLD. As the interannual nitrate anomalies averaged over the mixed layer are correlated with both the MLD interannual anomalies (Fig. 12b, 0.63 correlation) and the maximum absolute MLD (not shown; 0.60 correlation), it however not possible to discriminate between the two processes. In any case, the above results suggest that interannual winter chlorophyll variations largely result from phytoplankton growth through nutrient input to the MLD through turbulent processes. Although these results are consistent with Prasanna Kumar et al. (2001) hypothesis, it does not preclude Marra and Barber (2005) hypothesis to operate, i.e. a MLD control through a modulation of the grazing pressure.

Fig. 12 also allows discussing the processes driving the interannual MLD variability in winter. There is a significant relationship between the modelled interannual MLD and net surface heat fluxes (Fig. 12c, -0.83 correlation) anomalies during winter. The typical spatial pattern of anomalous net heat flux displays a broad heat flux cooling over the entire northern AS, with maximum anomalies located at the northern end of the AS for both model and observation (Fig. 10d, h). This is consistent with a MLD deepening controlled by convective overturning, which in turn is controlled by surface heat fluxes. Interannual net heat flux variations in this region are strongly related to 2-m air temperature anomalies (Fig. 12d, correlation 0.63), indicative of southward advection of anomalously cold/warm air from the continent driving anomalous blooms, as already suggested by Keerthi et al. (2016). A significant relationship also exists between the modelled interannual MLD variability and sea surface temperature (SST) variability in winter (Fig. 12e). A deeper MLD is associated with a cooler SST (-0.69 correlation), which in turn is driven by net heat flux variability (Fig. 12f, 0.66 correlation). This finally results in a large -0.79 negative correlation between the SST and SChl winter interannual anomalies, because they are both initially driven by the same surface heat flux anomalies. Observations exhibit a similar correlation (-0.73), reinforcing the

above conclusions from the model results.

## 5. Summary and discussion

### 5.1. Summary

The AS is one of the most productive regions in the world ocean, with a strong monsoon-driven seasonal cycle in SChl. The largest SChl bloom occurs during the summer monsoon in the western AS, in response to coastal and offshore upwelling driven by the Findlater jet. There is however also a prominent SChl bloom in winter in the northern AS, which exhibits large year-to-year fluctuations in its extent and intensity. These variations have not yet been described in detail and there is no consensus on their driving mechanism. In this paper, we described the NAS winter bloom interannual variability and the mechanism driving this variability. To reach that goal, we combined the analysis of several observational datasets (remotely-sensed chlorophyll products from various satellites and physical oceanic parameters derived from Argo in-situ profiles) and a biogeochemical model simulation.

Our results reveal that SChl anomalies from the various satellite products exhibit a good phase agreement, but large amplitude discrepancies. There is a strong (~ +/- 50% of the climatological value) year-to-year variability of the NAS winter SChl bloom. These fluctuations of the bloom amplitude are much better correlated (r~0.4 to 0.9 depending on the satellite product) to MLD than to TCD (r~-0.2 to 0.1) interannual fluctuations. As a result, correlations with MLD interannual anomalies are significant at the 90% confidence level in four out of six chlorophyll satellite products, but are insignificant for TCD anomalies irrespective of the product.

The above analysis is based on a limited number of years in observations, due to the in-situ data temporal coverage, which only becomes sufficient after 2002 thanks to Argo profilers. Using a biogeochemical model allows us extending our analyses over a longer period (1993-2012) and to analyse subsurface chlorophyll data (which are not available from observations). The model agrees well with observations in terms of both MLD and SChl winter interannual anomalies averaged over the NAS (typically r~0.7). As in observations, we find no relationship between the winter NAS SChl anomalies and the TCD anomalies in this simulation, contrary to what would be expected in Wiggert et al. (2002) mechanism. Rather, we find a strong relationship (r~0.6) between MLD and SChl anomalies, as in the observations (r~0.7). The analysis of the model vertical structure indicates that the increase in SChl is not the result of the upward mixing of a pre-existing subsurface chlorophyll maximum. Rather, enhanced surface heat losses due to the advection of cold air by northerly winds result in a more convective overturning, an anomalously deep seasonal MLD and more turbulent fluxes of nutrients into the MLD. This promotes new production in the surface layer. Our study therefore demonstrates that the mechanisms controlling chlorophyll variations at seasonal timescales (Prasanna Kumar et al., 2001; Lévy et al., 2007, Koné et al., 2009)

also operate at the interannual timescale. Despite these convincing evidences on the dominant role played by MLD variations in driving the year-to-year fluctuations of the winter NAS bloom, other oceanic processes such as the Ekman pumping or offshore advection could also play some role. Performing similar composite analyses as those displayed on Fig. 9 but for wind stress curl and surface currents anomalies does not reveal any significant relationship between these variables and winter interannual chlorophyll variations (not shown), suggesting that Ekman pumping or the advection of chlorophyll and/or nutrient is unlikely to play a strong role on the interannual fluctuations of the winter bloom in the NAS.

### 5.2. Discussion

The present study hence brings new insights on the interannual variability of the NAS winter bloom. As described in the introduction, there have indeed been to date only two studies addressing the physical mechanisms controlling this interannual variability (Prasanna Kumar et al., 2001; Wiggert et al., 2002), which proposed different mechanisms relying on the analysis of a very limited number of winters. Our study allows demonstrating which mechanism dominates based on the analysis of much longer and various datasets (12 winters in observations and 20 winters in the model). Our observational and modelling results are both inconsistent with the hypothesis proposed by Wiggert et al. (2002), i.e. that the TCD in winter controls the bloom amplitude through a daily dilution effect. In contrast, it is consistent with the hypothesis that interannual MLD variations largely control the amplitude of the bloom (Prasanna Kumar et al., 2001; Marra and Barber, 2005).

On the observational front, our results rely on a comparison of satellite-derived SChl interannual variations with in-situ derived MLD and TCD variations. Those datasets are both subject to uncertainties arising from both measurements accuracy and sampling issues (i.e. the data density used to derive interannual NAS-averaged anomalies). The satellite-derived SChl interannual variations are likely to be very robust as the different satellite products in the NAS exhibit a very good phase agreement (Fig. 4) and a very good data coverage during the winter season (Table 1). Regarding the in-situ derived MLD and TCD products, the large number of individual measurements used to build seasonal anomalies (i.e. from 100 to 500 depending on the years considered) yields a good accuracy on the estimate of these seasonal anomalies (~2 m uncertainty on MLD and ~3m on TCD estimated from a Monte Carlo approach by subsampling available data, is relatively small compared to the ~20 m peak to peak amplitude of observed interannual variations of those fields). One of the major limitation of this in-situ dataset may hence only be the inhomogeneous spatial sampling in the region considered, with a higher data density along a shipping line crossing the NAS box. The very good agreement between the in-situ data and the totally independent model (for which average are obtained over the entire NAS region, Fig. 7) however suggests that this is not the case. The fact that similar conclusions can be drawn from the model and the independent in-situ dataset also strengthens the trust in each of those datasets.

The diurnal cycle plays a key role in the hypothesis proposed by Wiggert et al. (2002). Although the spatio-temporal coverage of our in-situ dataset is sufficient to accurately sample MLD interannual variations, it does not allow monitoring the diurnal MLD variability. A proper investigation of the impact of diurnal variability on interannual chlorophyll variations from observations would hence require continuous and long-term temperature and chlorophyll profiles from a fixed location, which are not available to date. If Wiggert et al. (2002) mechanism was dominating, there should however be a negative correlation between interannual variations of chlorophyll and TCD (i.e. a deeper thermocline leading to lower chlorophyll concentrations operating through daily dilution). Table 2 and Figs. 8, 9, 10 clearly demonstrate that it is not the case. In addition, despite the absence of a diurnal cycle in the model forcing (i.e. by construction, the model can't reproduce the Wiggert et al. (2002) mechanism), the model displays a good agreement with observed interannual chlorophyll variability in winter in the NAS (see Figs. 7a and 10), which is an indirect evidence that the bulk of interannual chlorophyll variations are not linked to a modulation of night-time penetration of diurnal mixing by the TCD.

Wiggert et al. (2005) further argued that the inconsistency between Prasanna Kumar et al. (2001) and Wiggert et al. (2002) may be due to the different seasonal window considered in the two studies: December-January for Wiggert et al. (2002) and February for Prasanna Kumar et al. (2001). Wiggert et al. (2005) argued that during these two periods, two distinct processes drive the phytoplankton growth. On one hand, the transition from the winter monsoon to the spring inter-monsoon is characterized by detrainment blooms stimulated by increased irradiance received by phytoplankton due to mixed layer shoaling that follows the relaxation of monsoon winds. On the other hand, the beginning of the northeast monsoon is characterized by entrainment blooms that are stimulated by an increase in nutrients resulting from a deepening of the mixed layer in that period. To revisit this argument, we repeated our analysis over these two periods (Table 3). Our analysis indicates that interannual SChl anomalies during the beginning of the winter monsoon period do not exhibit any significant relationship neither with MLD nor with TCD anomalies (correlation below 0.30 not significant at the 90% significance level), i.e. neither Wiggert et al. (2002) mechanism nor Prasanna Kumar et al. (2001) mechanism is at work during the bloom initiation period. However, a significant relationship exists between SChl anomalies and MLD anomalies during post winter period (February-March) with a correlation of 0.80 in the observations and 0.51 for models (Table 3), suggesting that interannual MLD variations control the amplitude of the bloom during the period of peak interannual variability in winter (February-March).

Even though MLD and SChl interannual variations co-vary for most winters, the winters of 2002 and 2004 behave inconsistently relative to other years in both the model and observations (Fig. 7ab and 8a), suggesting that another mechanism could be at work during these years. Banerjee and Prasanna Kumar (2014) demonstrated that episodic dust storms could contribute to the interannual variability of the winter bloom in the central AS, away from the region of active winter convection. However, no episodic dust storms were reported during winters of 2002 and 2004 (Banerjee and Prasanna Kumar, 2014), indicating that iron inputs from dust storms are not responsible for the peculiar behaviour observed during

these two winters. In any case, our simulation captures most of the observed interannual variability of the chlorophyll bloom in winter (Fig. 7a) despite the use of a climatological iron aerial deposition forcing (i.e. iron deposition interannual variations are not accounted for). This suggests that dust storms interannual variations do not play a dominant role in driving the interannual variability of the bloom in the northern AS. The apparent contradiction between our results and those of Banerjee

and Prasanna Kumar (2014) may arise from the different regional focus: the northern AS where winter convection occurs for our study, and the central AS where no convective overturning occurs for Banerjee and Prasanna Kumar (2014). The different vertical physics in the two regions may imply a different role of micronutrients.

Our results point toward a strong control of the MLD variability in the NAS through anomalous heat flux

perturbations. Although we generally refer to interannual MLD variability in this study, the largest MLD fluctuations in NAS are observed over a single month (see Fig. 7b). These variations hence rather occur at intraseasonal timescales but translate into interannual anomalies when averaging over the entire winter season. Keerthi et al. (2016) already provided a detailed description of these intraseasonal MLD fluctuations in this region in winter, relating them to the advection of continental temperature anomalies from the northern end of the basin. The climate variability behind these heat flux

fluctuations is however currently unknown. Wiggert et al. (2009) results point to a contrasted biological signature of the western AS during the 1997/1998 and 2006/2007 El Niño events, with an overall decrease of productivity during the earlier and a slight increase during the latter. The extended analysis of a 1961-2001 model hindcast (Currie et al., 2013) indicates that El Niño generally results in an anomalously low winter and fall SChl over the AS (as seen in 1997-98), and a negligible impact of the IOD. In line with Currie et al. (2013), we find that interannual variations of the winter 2-m air temperature in

the NAS box over the 2002-2011 period are strongly correlated with ENSO (correlation=0.77) and weakly correlated with IOD (correlation=0.05), suggesting that the surface forcing that drives NAS MLD interannual variability may be ENSO-driven (Table 4). Regarding the impact on SChl, all observed and model chlorophyll products do exhibit a modest negative correlation between ENSO and interannual fluctuations of the winter bloom over the 2002-2011 period (Table 4). This is consistent with the hypothesis that an El Niño event drives a weaker winter monsoon, warmer surface air, less convective

overturning and hence a weaker bloom. The level of correlation is however modest and not statistically significant in all datasets (ranging from -0.11 to -0.44, depending on the SChl product), indicating that ENSO is not the only driver of the interannual winter bloom variations in this region. In addition, this influence of ENSO is not very stable in time (Table 4), with a larger correlation with 2-m air temperature over the recent 2002-2011 period (0.77) compared to the extended 1993-2011 period (0.27). Further detailed analyses are required to ascertain and better understand the influence of ENSO on the

AS winter monsoon. Finally, while interannual SChl variations are rather consistent amongst products, linear SChl trend are not consistent amongst products, some of them showing a decreasing trend and some others showing an increasing trend. These discrepancies hence prevented us to perform a robust assessment of these trends in the present study. We however performed the analyses in the paper using detrended data and found that our results regarding the interannual variability are robust.

An obvious perspective of this study is to investigate the processes controlling the interannual chlorophyll variations in summer, which are far larger than in winter. As compared to the winter season, an analysis based on observations during the summer monsoon is complicated by the poorer satellite data coverage (Table 1) and the fewer Argo profiles in the

5  Oman/Somalia upwelling region (Fig. 2b). The model analysis may however provide further insights on the mechanisms that control the upwelling productivity during summer. Long-term variations also deserve further analysis. By analysing a seven year-long satellite dataset, Goes et al. (2005) suggested that the southwest monsoon intensifies as a result of climate change, driving increased upwelling, primary production and ecosystem changes in the AS. However, the shortness and discontinuity of the data questions the reliability of these results (Beaulieu et al. 2013). Recent studies based on longer datasets (Roxy et

10  al. 2015, 2016) rather point towards a reduction of the summer monsoon winds and an AS summer bloom reduction due to enhanced upper ocean stratification in response to climate change. Climate change projections from coupled experiments however exhibit a large range of responses in terms of changes in the southwest monsoon (e.g. Turner and Annamalai, 2012), and of productivity in the AS (e.g. Bopp et al., 2013). Those large uncertainties call for more targeted studies of the impact of climate change on oceanic productivity in the AS.

**Acknowledgments:**

This study was supported by the Centre National d'Etudes Spatiales (CNES) CLIMCOLOR project. Keerthi M. G is funded by the IFCPAR (Indo French Centre for Promotion of Advanced Research) proposal 4907-1. Model experiments were performed at HPC Pravah at CSIR-NIO. M. Lengaigne, C. Ethé, J. Vialard, M. Levy, de Boyer Montegut and O. Aumont are

25  funded by Institut de Recherche pour le Développement (IRD). Parvathi V is funded by CSIR under SRF. I. Suresh and Akhil V. P. acknowledge the financial support from CSIR, New Delhi. We thank Goddard GSFC/NASA for providing the SeaWiFs, MERIS and MODIS chlorophyll data, ESA-GLOBCOLOUR for providing the GSM and AVW chlorophyll data and ESA-OCEANCOLOUR for providing the OC-CCI chlorophyll data. This has NIO contribution number 6068.

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

| Name | Satellite | Algorithm | Period | Grid | Winter/Summer NAS coverage |
|---|---|---|---|---|---|
| SeaWiFS | GeoEye OrbView-2 | OC4v6 (0'Reily et al, 2000) | 1997- 2010 | 9km | 92% / 25% |
| MERIS | ESA ENVISAT | OC4Me (0'Reily et al, 2000) | 2002-2012 | 9km | 96% / 30% |
| MODIS | Terra and Aqua | OC3v5 (0'Reily et al, 2000) | 2002-present | 9km | 96% / 20% |
| GSM | Combine SeaWiFs/MODIS/MERIS | GSM (Maritorena and Siegel, 2005) | 1997- 2012 | 4km | 98% / 45% |
| AVW | Combine SeaWiFs/MODIS/MERIS | | 2002-2012 | 4km | 98% / 53% |
| OC-CCI | Combine SeaWiFs/MODIS/MERIS | OC-CCI-v2.0 (Grant et al 2015) | 1997- 2013 | 4km | 98% / 58% |

**Table 1:** Main characteristics of ocean color satellite products used in the present study

| SChla Data | Cor (MLDa) | Cor (TCDa) |
|---|---|---|
| SeaWiFS (2003-2010) | 0.46 | -0.006 |
| MERIS (2003-2012) | **0.69** | 0.05 |
| MODIS (2003-2012) | **0.86** | -0.02 |
| OC-CCI (2003-2012) | **0.72** | 0.05 |
| GSM (2003-2012) | **0.77** | -0.19 |
| AVW (2003-2012) | 0.38 | -0.17 |

**Table 2:** Correlation between average NAS box winter (DJFM) SChl interannual anomalies derived from satellite products and in-situ MLD and TCD interannual anomalies. Bold typeface indicates correlations which are statistically different from zero at the 90% confidence level.

| | | December-March (DJFM) | December-January (DJ) | February-March (FM) |
|---|---|---|---|---|
| **Correlation** | | | | |
| | **MLDa Vs SChla** | **0.72** | -0.05 | **0.80** |
| **OC-CCI** | **TCDa Vs SChla** | 0.05 | 0.43 | -0.10 |
| **Model** | **MLDa Vs SChla** | **0.59** | 0.21 | **0.51** |
| | **TCDa Vs SChla** | 0.3 | 0.37 | 0.19 |

**Table 3:** Correlation between SChl interannual anomalies and MLD and TCD interannual anomalies averaged over the NAS box for December-March, December-January and February-March. Bold typeface indicates correlations which are statistically different from zero at the 90% confidence level.

| | Correlation | |
|---|---|---|
| | **IOD (SON)** | **ENSO (NDJ)** |
| **T2a (1993-2011)** | -0.06 | 0.27 |
| **T2a (2002-2011)** | 0.05 | **0.77** |
| **SChl_OC-CCI (2002-2011)** | 0.07 | -0.34 |
| **SChl_MODIS (2002-2011)** | 0.26 | -0.39 |
| **SChl_MERIS (2002-2011)** | 0.04 | -0.37 |
| **SChl_GSM (2002-2011)** | 0.24 | -0.44 |
| **SChl_AVW (2002-2011)** | 0.05 | -0.11 |
| **SChl_Model (2002-2011)** | 0.02 | -0.30 |

**Table 4:** Correlation of IOD and ENSO index with NAS box averaged winter (DJFM) 2m-surface temperature interannual anomalies (T2a) and SChl interannual anomalies derived from different satellite products and model. Bold typeface indicates correlations which are statistically different from zero at the 90% confidence level.

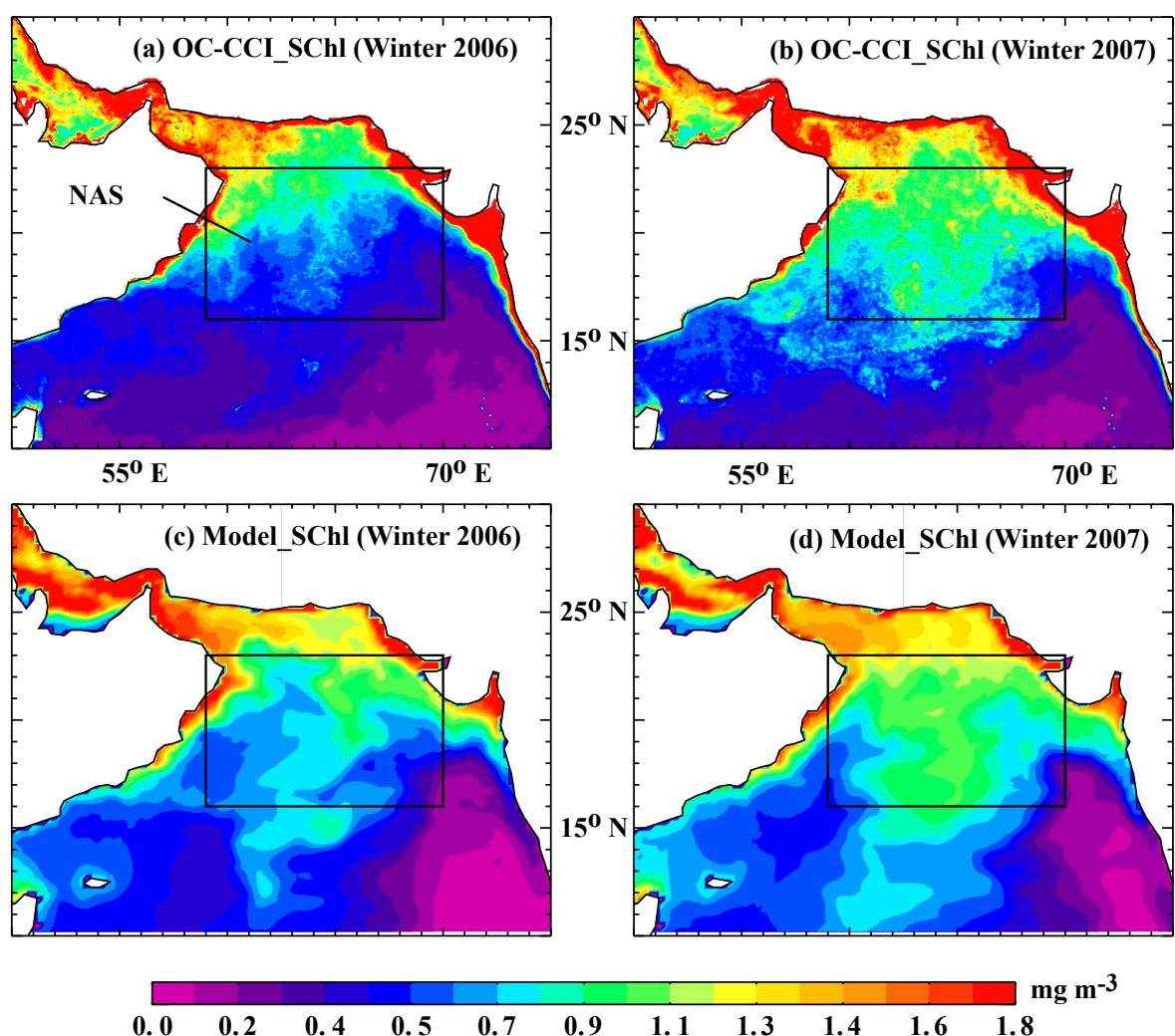

**Figure 1:** Arabian Sea average SChl for the winter (DJFM) of **(left panels)** 2006 and **(right panels)** 2007 in **(top)** OC-CCI product and **(bottom)** model. The NAS (North Arabian Sea) box (59°E-70°E, 16°N-23°N) is indicated by a black frame for future reference.

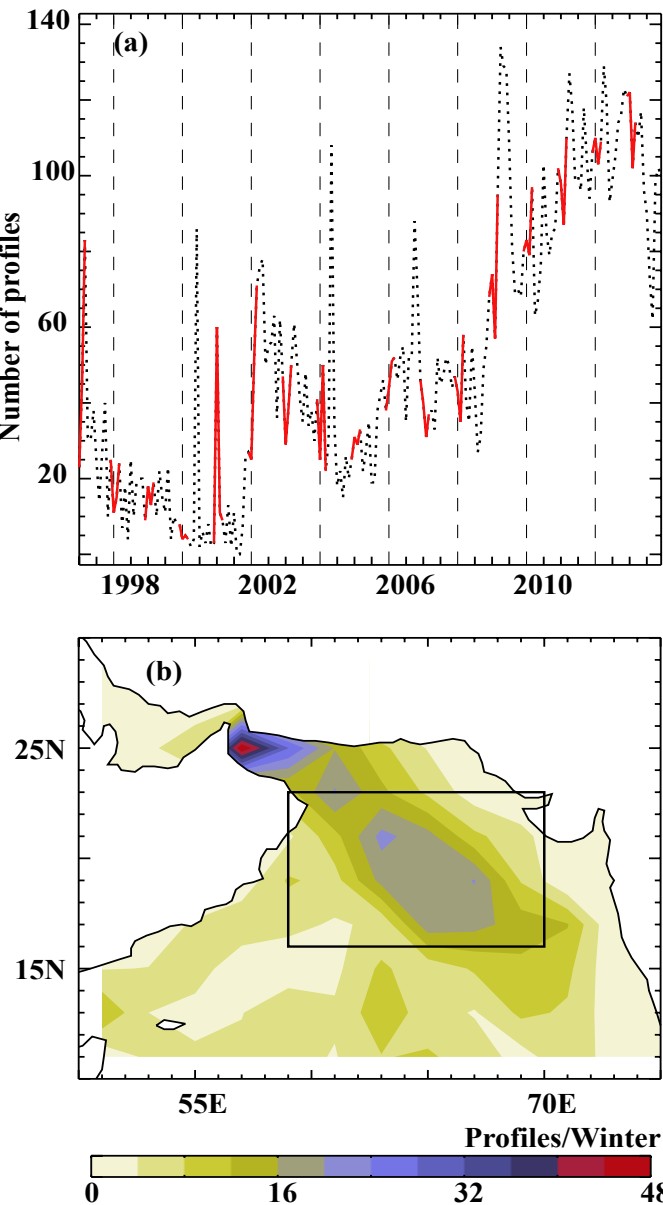

**Figure 2: (a)** Time series of the number of in-situ profiles per month over the NAS box, from 1997 to 2013. The curve is highlighted in red for winter (DJFM). **(b)** Average winter in-situ profiles density (per 2°x2° box and per season) for 2002-2012. The NAS region is indicated by a black frame on panel b.

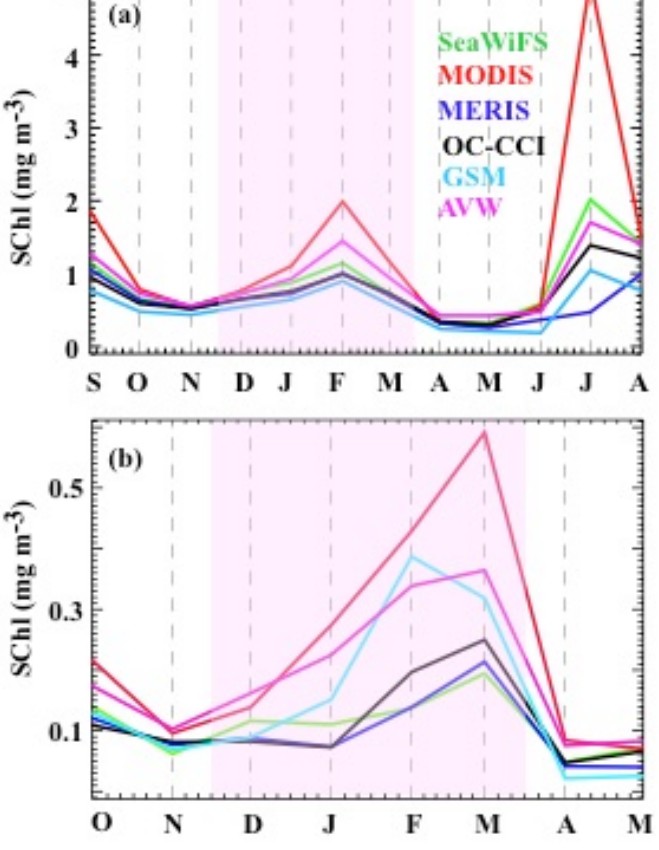

**Figure 3. (a)** Climatological monthly seasonal cycle of NAS-average SChl for all satellite products. **(b)** Standard deviation of monthly (October-May) NAS-averaged SChl for all satellite products.

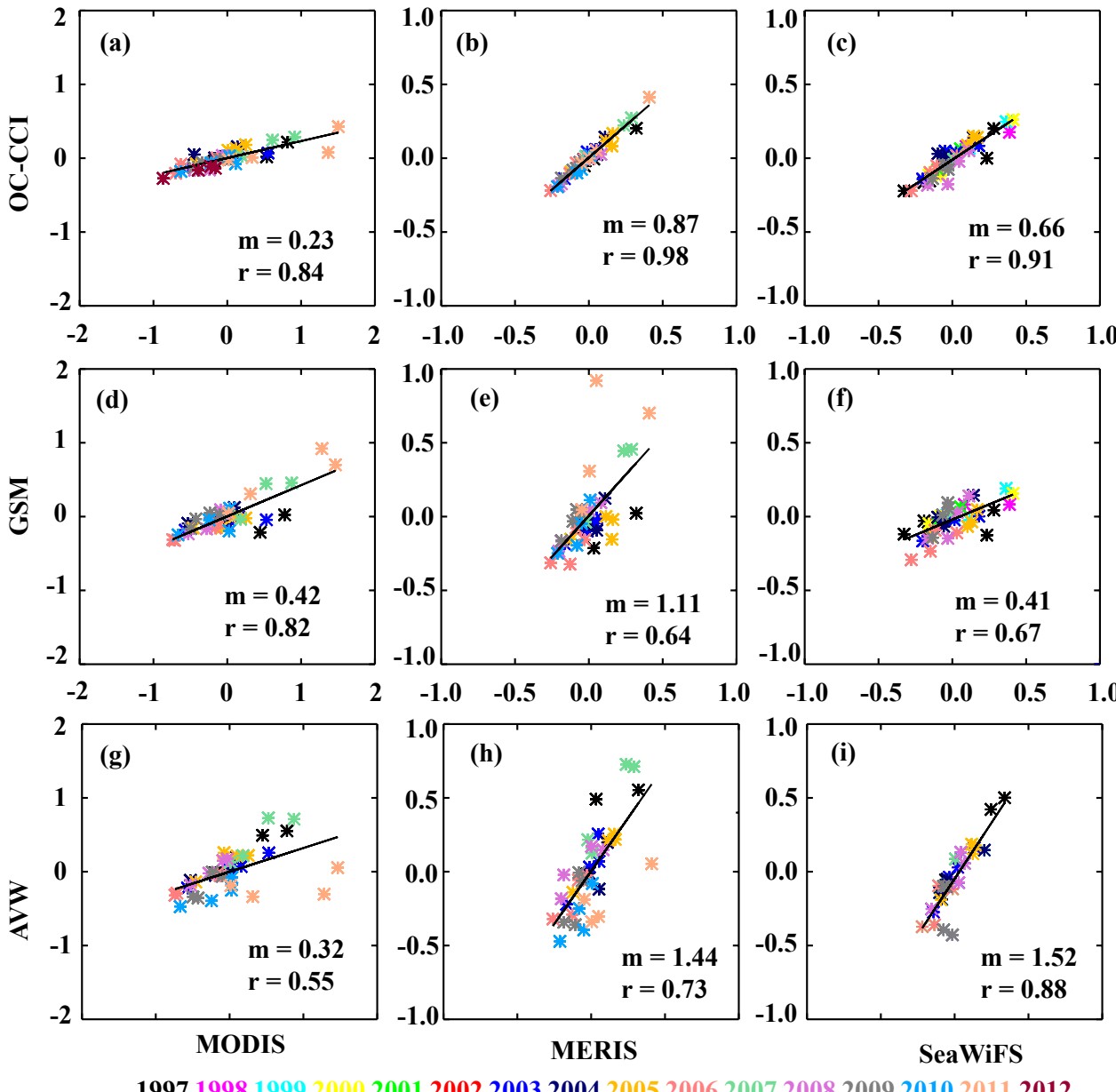

**Figure 4. (First row)** Scatter plot of NAS-average winter (DJFM) monthly SChl interannual anomalies (mg.m$^{-3}$) in OC-CCI product against MODIS, MERIS and SeaWiFS products. **(Second row)** Idem for GSM product. **(Third row)** Idem for AVW product. All the correlations (r) and regression coefficients (m) indicated on each panel are significantly different from zero at the 90% confidence level.

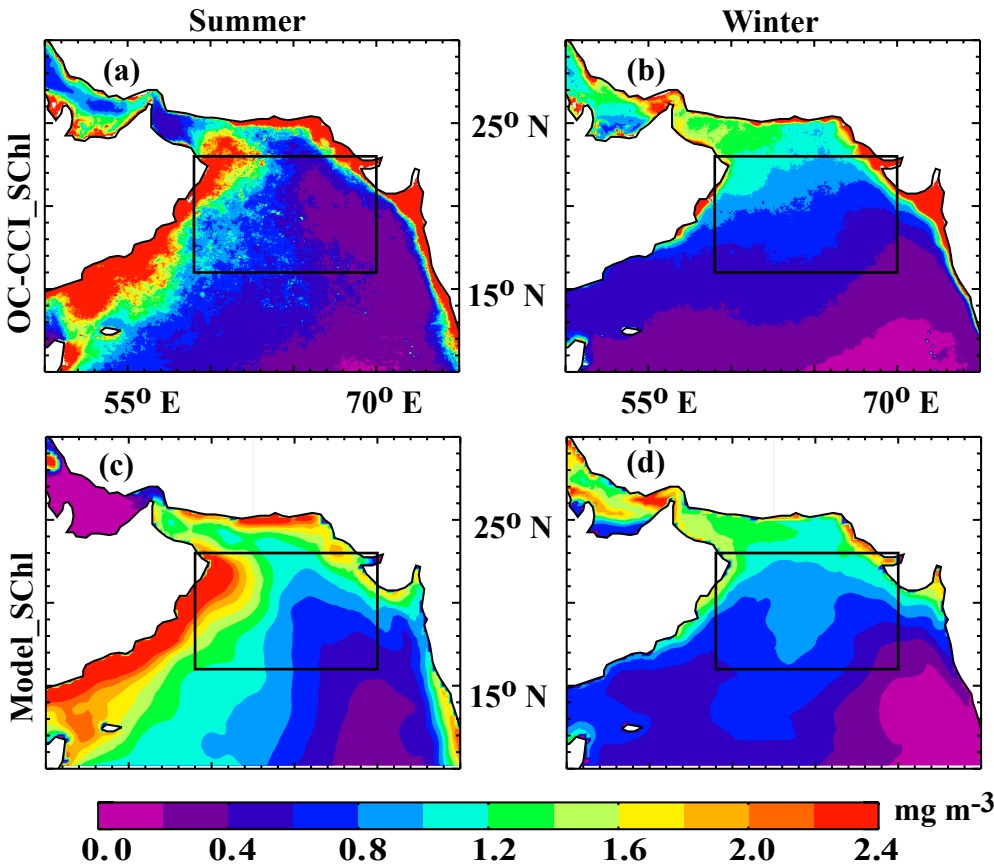

**Figure 5:** Arabian Sea climatology of (left panels) summer (JJAS) and (right panels) winter (DJFM) **(top)** OC-CCI and **(bottom)** modelled SChl (mg.m⁻³).

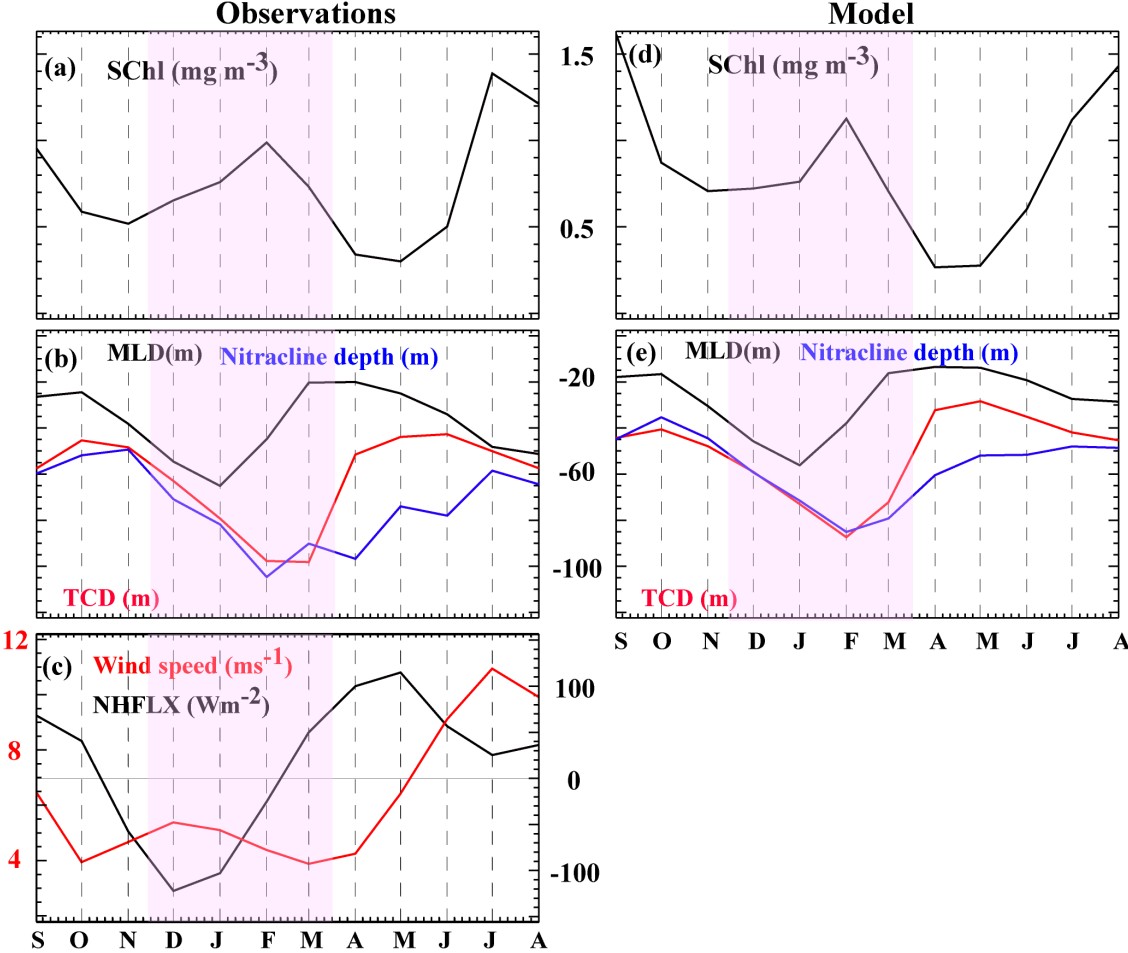

**Figure 6:** Mean seasonal cycle of NAS box-averaged monthly **(a)** SChl, **(b)** MLD (black line), TCD (red line) and nitracline depth (blue line), **(c)** surface net heat flux (NHFLX-black line) and wind speed (red line) in observations. **(d-f)** Same for model. The NAS box is outlined on Figure 1. Note that the model average is based on the entire NAS box, while observations subsample this box: the good agreement between the two averages however suggest that observational subsampling does not introduce large biases.

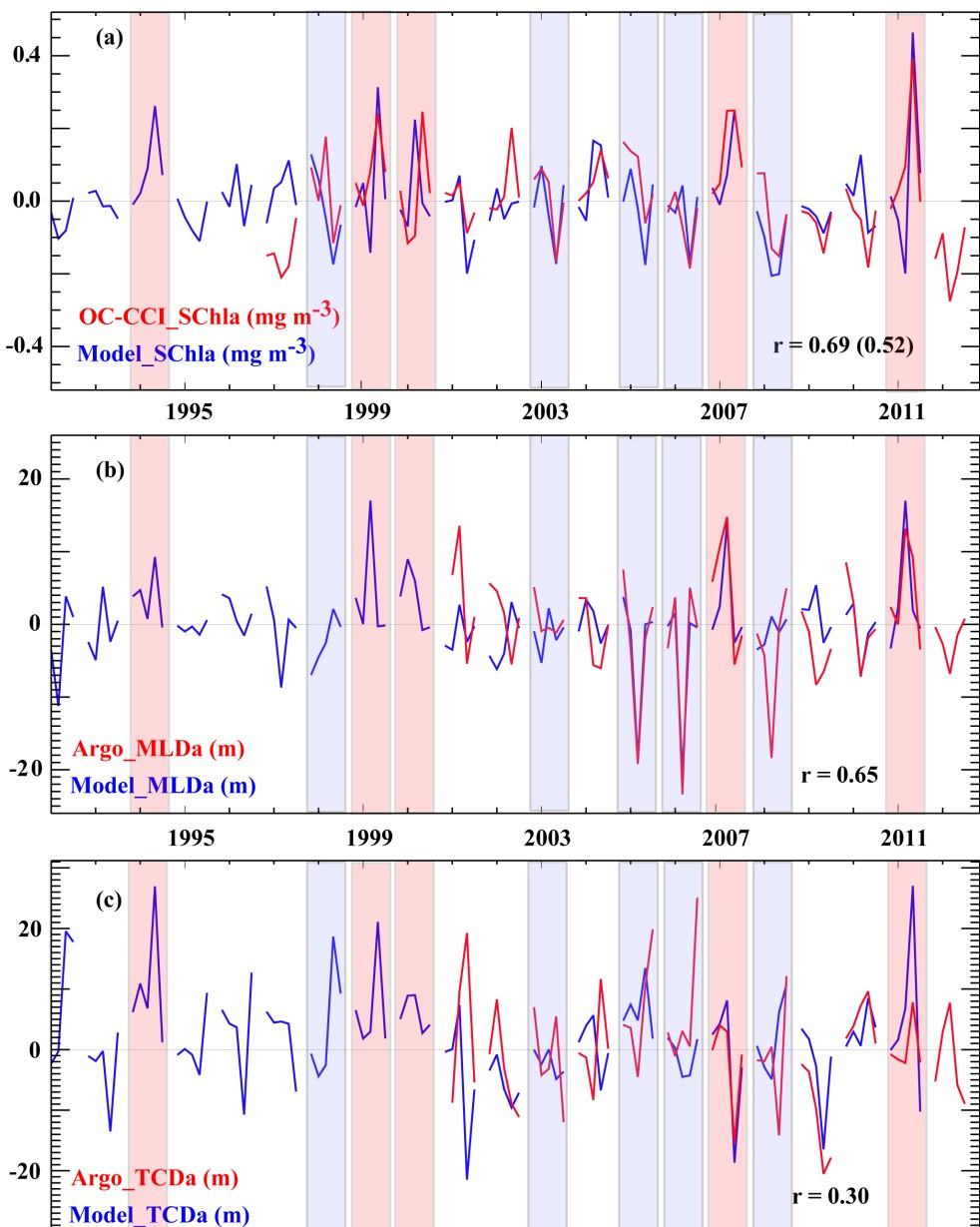

**Figure 7:** Monthly time series of NAS box-averaged modelled and observed interannual anomalies in winter for **(a)** SChl, **(b)** MLD and **(c)** TCD. The correlation between the model and observations from 2002-2012 is indicated on panels (a-c). Red and blue shadings respectively indicate winters for strong and weak blooms in the model, considered for the composite plots of Figure 10. Note that the model average is based on the entire NAS box, while observations subsample this box: the good agreement between the two averages however suggest that observational subsampling does not introduce large biases.

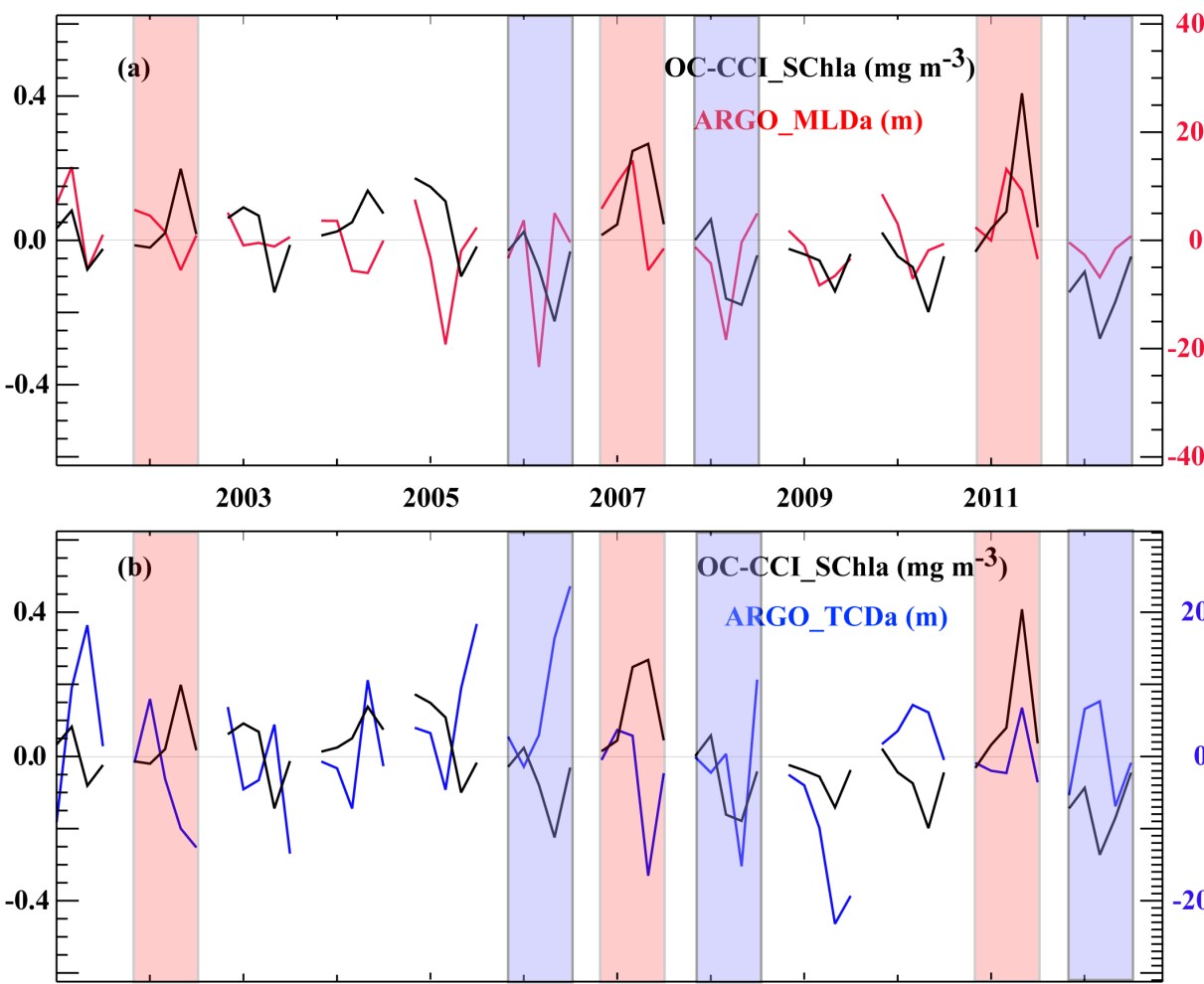

**Figure 8:** Observed monthly time series of box-averaged **(a)** SChl and MLD and **(b)** SChl and TCD anomalies in winter over the 2002-2013 period. The red (blue) shadings highlight the winters of strong (weak) blooms used in the composite plots of Figure 10.

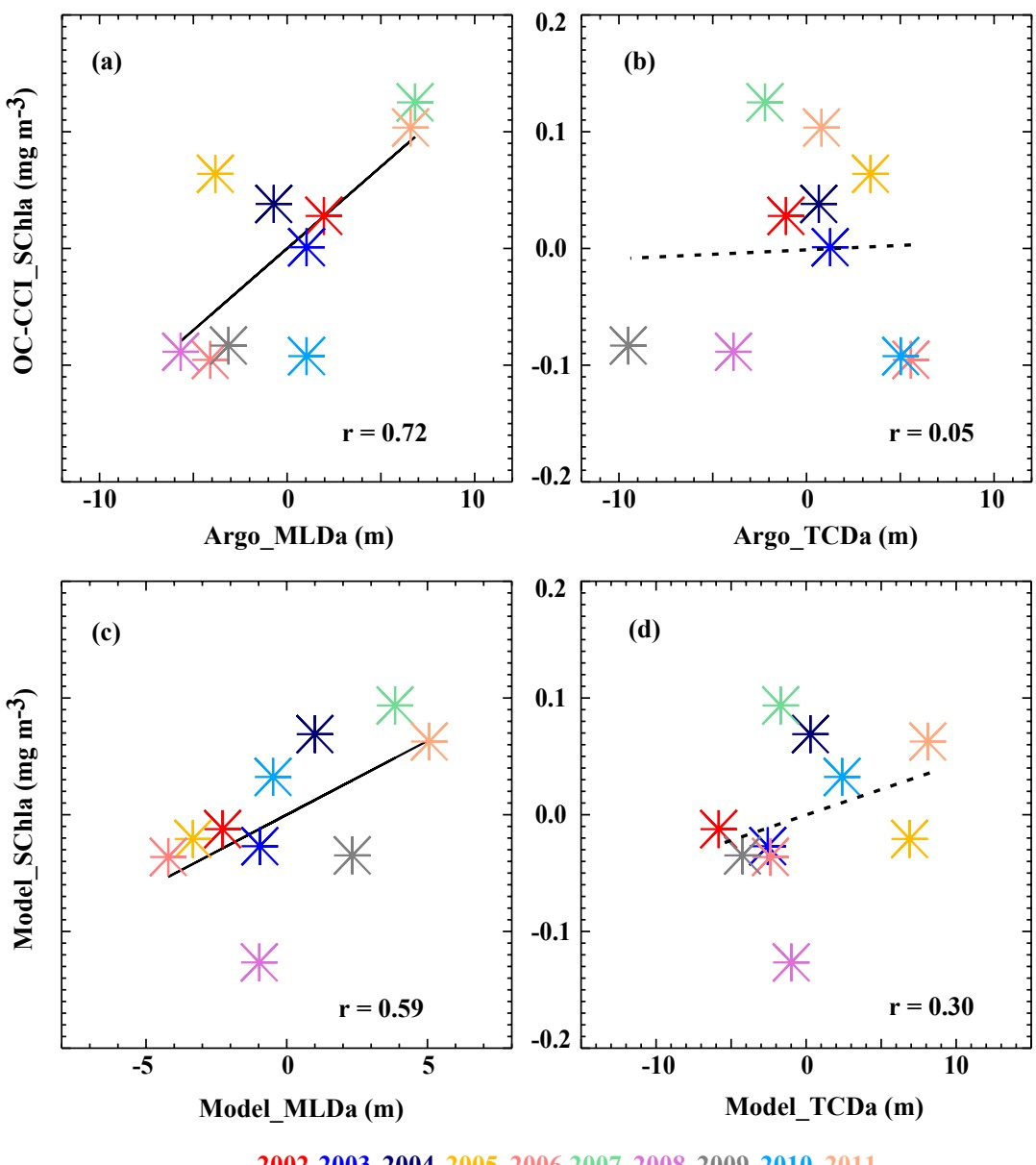

**Figure 9:** Scatterplot of winter NAS-averaged OC-CCI SChl anomalies against observed **(a)** MLD and **(b)** TCD anomalies over the 2002-2011 period. **(c-d)** Idem for model. Plain (dashed) lines indicate the linear regression that is (not) significantly different from zero at the 90% confidence level.

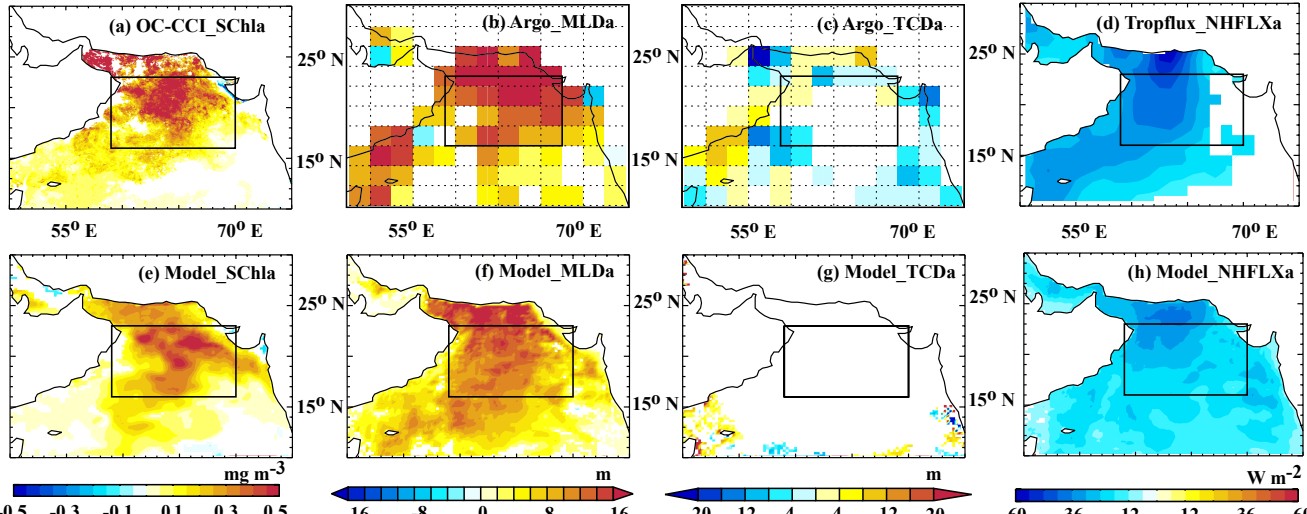

**Figure 10:** Observed interannual anomalies of **(a)** SChl (OC-CCI), **(b)** MLD and **(c)** TCD (Argo-derived) **(d)** net surface heat flux (Tropflux) for composite SChl blooms (build from half of the difference between positive and negative events highlighted on Figure 8). The SChl composites are built from the months of max SChl anomaly: March 2003, March 2008 and March 2012 for positive events; March 2007, March 2009 and February 2013 for negative events. The MLD and TCD composites are built from the months of February of the same year). **(e-h)** Idem for the model. For the model, composites are built from the positive and negative events highlighted on Figure 7 (SChl is composited using March 1995, 2000, 2008, 2012 and February 2001 for positive events and March 1999, 2004, 2006, 2007, 2009 for negative events; MLD and TCD are composited using February of the same years). Regions where composite values are less than the standard error are displayed in white.

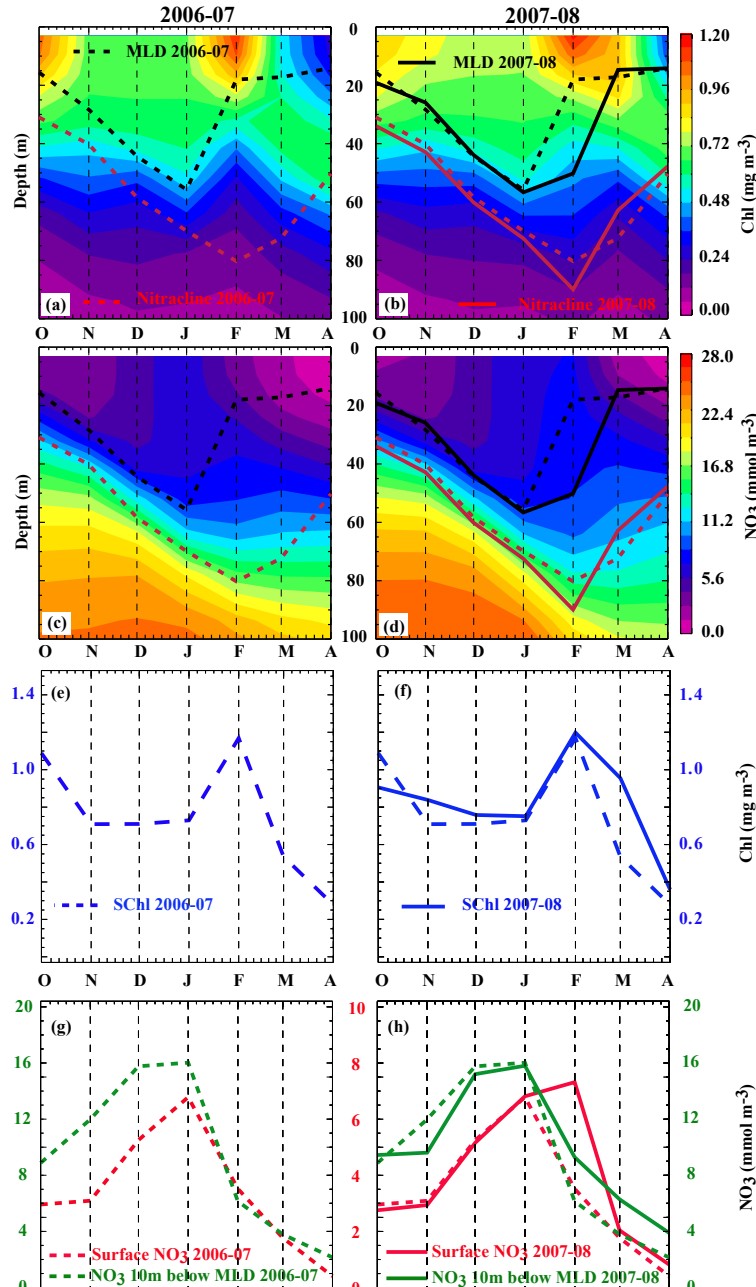

**Figure 11:** Depth-Time section of NAS-averaged **(a, b)** chlorophyll (Chl) and **(c, d)** nitrate ($NO_3$) for (a, c) 2006-07 and (b, d) 2007-08. The black lines indicate the MLD, in thick for 2007-08 and dashed for 2006-07. The red lines similarly indicate the nitracline depth. Time series of NAS-averaged **(e, f)** SChl (blue curve) and **(g, h)** surface nitrate (red curve) and nitrate concentration 10m below the bottom of the MLD (green curve) for (e ,g) 2006-07 and (f, h) 2007-08. On panels f and h, the 2006-07 values have been reported as dashed curves to ease comparisons between the two years.

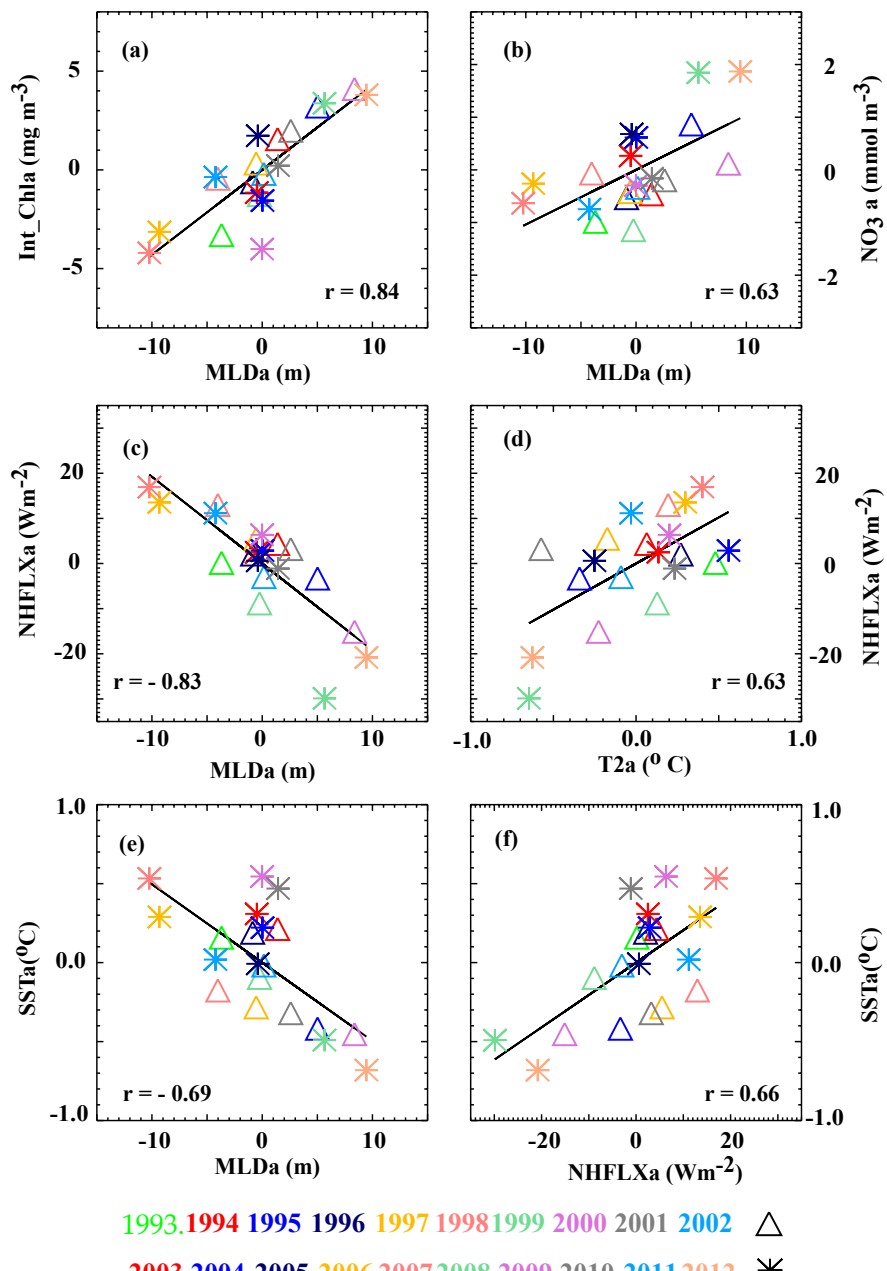

**Figure 12:** Scatterplot of modelled NAS box-averaged winter (February-March) interannual anomalies of **(a)** 0-200m total chlorophyll content against MLD. **(b)** Average MLD nitrate vs MLD. **(c)** surface net heat flux vs MLD. **(d)** surface net heat flux vs 2m air temperature. **(e)** SST vs. MLD. **(f)** SST vs. surface net heat flux. Plain lines indicate regression coefficients that are significantly different from zero at the 90% confidence level.

