# Peer review of "Physical control of the northern Arabian Sea winter chlorophyll bloom interannual variations"

_Biogeosciences, 2016_

## Referee Comment (RC1) · Anonymous Referee #1 · 5 Jul 2016

General Comments:

The Arabian Sea is one of the most productive regions of the world oceans which experiences phytoplankton blooms during boreal summer and winter. Both these blooms are reasonably well characterized and the physics controlling them are reasonably well studied. However, what is still largely least explored is the inter-annual variability of the summer as well as the winter blooms. The major reason why it has not yet been adequately addressed is the limitation in long time-series data. It is in this context that the present study assumes importance.

The present paper addresses the inter-annual variability of the winter chlorophyll bloom in the northern Arabian Sea (NAS) using both observation and simulations from a

coupled biophysical ocean model. Towards the observational data, the authors used satellite-derived chlorophyll pigment concentrations, and Argo-derived mixed layer and thermocline depths. Using these, the authors aim at "better understanding the inter-annual variability of the NAS winter bloom" (lines 15).

The central theme of the study is the processes that leads to the phytoplankton bloom in NAS during winter and to ascertain whether the process suggested by Prasanna Kumar et al. (2001) or that by Wiggert et al. (2002) explains the bloom. The authors conclude that the winter blooms are strongly tied to mixed layer depth and the resulting modulation of nutrient entrainment into the mixed layer, a result similar to that of Prasanna Kumar et al. (2001). The inter-annual variability of NAS winter bloom amplitudes are controlled by the variation in the net heat flux at the air-sea interface, which in turn controls the MLD and nutrient entrainment.

It is a well written manuscript and should be published, in my opinion, but only after consideration of some of the comments listed below.

Major concerns: 1. What is the basis on which the authors use the months from January to April to define winter? This is not true in the case of the Arabian Sea and hence not acceptable. Let me explain.

Based on the mean seasonal cycle of net heat flux both from observation as well as model presented in Figure 6 (c & f), the ocean looses heat from November until February. From March onwards the ocean starts gaining the heat and the net heat flux remains positive until October. Note that in April ocean gains heat as much as 100 w/m2 indicating the warming of the ocean rather than the prevalence of winter conditions. So from the net heat flux point of view November to February defines the winter condition.

Similarly, the mean seasonal cycle of surface chlorophyll from observation and model in figure 6 (a &d)) shows an increase from November, peaks in February and returns to the November value by March.

In view of the above, November until March could be considered as winter while dealing with chlorophyll response in the box under study.

2. It has been shown in recent years that episodic dust storms that occur during winter are important in driving the interannual variability of chlorophyll in the Arabian Sea through the atmospheric input of nutrients, especially iron. See for example the studies of Wiggert and Murtugudde (2007), Patra et al (2007), Naqvi et al (2010) and Banerjee and Prasanna Kumar (2014). The authors need to at least address the role of dust-induced Fe input in driving the inter-annual variability of chlorophyll in their study region.

Minor concerns:

3. Page 7 Line 15 " The simulation captures the surface chlorophyll seasonal cycle in the NAS….."

While it is so during winter (even in winter note that the model SChl does not capture the increase from November to January seen in the observation), the model completely misses the declining SChl trend from July to August. Instead model depicts the continuation of monotonic increase from May to August.

Authors need to point this out while discussing the simulation results. 4. Page 8 lines 7-9 "The figure illustrates that the observed interannual SChl…."

The authors need to explain in the text/discussion what the inverse relationship between SChla and MLDa during 2003 & 2005 means.

5. Page 9 lines 2-3 "As a result, the mixed layer nitrate….."

Though the MLD nitrate concentration during 2008 Feb is double than that of 2007, the Chlorophyll did not show a commensurate increase. The authors need to explain this in the discussion in the of Redfield ratio and carbon fixation.

---

## Referee Comment (RC2) · Anonymous Referee #2 · 7 Nov 2016

General Comments:

The study, using multiple datasets and a model, examines the processes related to the interannual variability of winter chlorophyll in the northern Arabian Sea. The Arabian Sea, especially the coastal-open ocean upwelling regions are subjected to an interplay of oceanic processes, such as coastal upwelling, Ekman pumping, mixing, entrainment and lateral/vertical advection (Vialard et al 2012). Other than these physical processes, changes in light penetration and nutrient supply also regulate the chlorophyll content in this basin. Hence it is an interesting task to investigate the biophysical interactions over this region, which result in the observed chlorophyll variations. However, other than presenting several correlation analyses between the mixed layer depth and

chlorophyll variations, the study does not delve in depth into the processes governing the chlorophyll changes. The paper is neatly written but it is difficult for the reviewer to comprehend what the prime objectives of the study are, or if there is anything novel in the results. Due to these shortcomings, and as detailed below, I do not recommend the manuscript for publication at its current state.

Specific Comments:

1. The study revolves around a hypothesis put forward by Wiggert et al (2002) that the interannual variations of chlorophyll intensity is regulated by diurnal mixing. The current study, using correlation analyses, points out that the winter chlorophyll variability is tied to the mixed layer depth anomalies, which are associated with the surface heat flux anomalies. However, this factor has already been pointed by studies like Prasanna Kumar et al (2002), where they point out that surface cooling (due to evaporation) in the northern Arabian Sea, combined with reduced incoming solar radiation and high salinity, drives convective mixing, resulting in the upward transport of nutrients in the mixed layer. Similar to the current study, this study also compares two anomalous years and describe the processes involved.

2. Also, apart from the correlation analyses between mixed layer depth (and nutrients) and chlorophyll concentrations, the manuscript does not examine how the oceanic processes such as the Ekman pumping, offshore advection etc plays a role on the interannual variability. Also, to counter the study by Wiggert et al (2002), the current study does not examine the contributions due to diurnal mixing. In fact, it is not clear to me where Wiggert et al (2012) says that mixed layer variations are not important in controlling the chlorophyll concentrations – though the study says that diurnal changes of mixed layer are important.

3. Page 10, Line 10 says the MLD deepening is controlled by convective overturning, which in turn is controlled by surface heat fluxes. It is not clear to me how the cause and effect is separated here. Surface net heat flux is inversely proportional to the mixed

layer depth. Hence it is not surprising as in Fig.10 that they show a good correlation.

4. Page 12, Line 20 suggests that the interannual variations in the surface flux are modulated by ENSO (strong correlations). The connection with ENSO was shown by Murtuggude et al (1999), which examined the chlorophyll changes in the Indian Ocean with respect to the 1997-1998 El Niño and the 1998 La Niña. The El Niño – La Niña episodes were accompanied by changes in chlorophyll over the Arabian Sea, with low Chl concentrations during the El Niño period, followed by anomalously high concentrations during the La Niña episode. These changes were attributed to local ocean-atmospheric dynamics linked to the shifts in the Walker circulation.

On a similar case, the authors compare 2007 and 2008 MLD and chlorophyll. 2006/7 was a weak El Niño year and 2007/8 was a La Niña year, which is clearly reflected in the Chl anomalies, with the former resulting in negative anomalies and the latter in positive anomalies. Going by correlations as in the current study, I can say that ENSO is a major component in driving both the surface flux and chlorophyll anomalies in the Arabian Sea, on interannual timescales.

5. Also, how do the changes in Eurasian winds (Page 13, Line 8, Goes et al 2005) compare with the ENSO impact? Are the winds increasing, and do they have an impact on the Chl in the Arabian Sea? If so, how does it ride on the interannual variability imposed by ENSO?

References:

Murtugudde, R. G., Signorini, S. R., Christian, J. R., Busalacchi, A. J., McClain, C. R., & Picaut, J. (1999). Ocean color variability of the tropical Indo‐Pacific basin observed by SeaWiFS during 1997–1998. Journal of Geophysical Research: Oceans, 104(C8), 18351-18366.

Vialard, J., Jayakumar, A., Gnanaseelan, C., Lengaigne, M., Sengupta, D., & Goswami, B. N. (2012). Processes of 30–90 days sea surface temperature variability in the northern Indian Ocean during boreal summer. Climate dynamics, 38(9-10), 1901-1916.

---

## Author Response (AR1)

*General Comments:*

*The Arabian Sea is one of the most productive regions of the world oceans which experiences phytoplankton blooms during boreal summer and winter. Both these blooms are reasonably well characterized and the physics controlling them are reasonably well studied. However, what is still largely least explored is the inter-annual variability of the summer as well as the winter blooms. The major reason why it has not yet been adequately addressed is the limitation in long time-series data. It is in this context that the present study assumes importance.*

*The present paper addresses the inter-annual variability of the winter chlorophyll bloom in the northern Arabian Sea (NAS) using both observation and simulations from a coupled biophysical ocean model. Towards the observational data, the authors used satellite-derived chlorophyll pigment concentrations, and Argo-derived mixed layer and thermocline depths. Using these, the authors aim at "better understanding the interannual variability of the NAS winter bloom" (lines 15).*

*The central theme of the study is the processes that leads to the phytoplankton bloom in NAS during winter and to ascertain whether the process suggested by Prasanna Kumar et al. (2001) or that by Wiggert et al. (2002) explains the bloom. The authors conclude that the winter blooms are strongly tied to mixed layer depth and the resulting modulation of nutrient entrainment into the mixed layer, a result similar to that of Prasanna Kumar et al. (2001). The inter-annual variability of NAS winter bloom amplitudes are controlled by the variation in the net heat flux at the air-sea interface, which in turn controls the MLD and nutrient entrainment.*

*It is a well-written manuscript and should be published, in my opinion, but only after consideration of some of the comments listed below.*

We thank the reviewer for his positive comments and for his inputs on the paper. We provide a detailed answer to each of the comments below.

*Major concerns: 1. What is the basis on which the authors use the months from January to April to define winter? This is not true in the case of the Arabian Sea and hence not acceptable. Let me explain. Based on the mean seasonal cycle of net heat flux both from observation as well as model presented in Figure 6 (c & f), the ocean looses heat from November until February. From March onwards the ocean starts gaining the heat and the net heat flux remains positive until October. Note that in April ocean gains heat as much as 100 w/m2 indicating the warming of the ocean rather than the prevalence of winter conditions. So*

*The referee comments are italicized.* Answers in regular typeface. Actions taken in red.

*from the net heat flux point of view November to February defines the winter condition. Similarly, the mean seasonal cycle of surface chlorophyll from observation and model in figure 6 (a & d) shows an increase from November, peaks in February and returns to the November value by March. In view of the above, November until March could be considered as winter while dealing with chlorophyll response in the box under study.*

We agree with the reviewer that the definition we used for winter monsoon was not properly justified. As mentioned by the reviewer, Figure 6 clearly shows that the climatological heat flux is negative from November to February, which also corresponds to the period when the mixed layer depth (MLD) is deepest. The peak of the bloom is delayed by about one month relative to the deepest MLD due to the time it takes for the biomass to grow, and large chlorophyll values are still found in March. To better assess the seasonality of the amplitude of the winter bloom interannual variations, we calculated the monthly standard deviation of the interannual chlorophyll variations for each observed dataset. This new analysis is now included as Figure 3b in the revised manuscript (shown as Figure R1-1 below). This Figure clearly shows that the months of November and April correspond to minimum in the amplitude of interannual chlorophyll variability for most datasets. Based on this finding, we hence define the winter season as the period from December to March where interannual chlorophyll variations are larger. This is particularly true for February-March where interannual variations reach clear maximum amplitude. All the figures and text have been revised accordingly using this new season definition. This change of the seasonal window used to define the winter monsoon however does not change the overall conclusions of our study. This is illustrated on Figure R1-2 (similar to Figure 9 of the submitted version but for the new seasonal window considered).

[Figure]

**Figure R1-1:** Monthly standard deviation of surface chlorophyll interannual anomalies for all satellite products.

[Figure]

**Figure R1-2:** Scatterplot of winter (DJFM), NAS-averaged OC-CCI surface chlorophyll anomalies against observed (a) MLD and (b) thermocline depth anomalies. (c-d) Idem for Model.

*2. It has been shown in recent years that episodic dust storms that occur during winter are important in driving the interannual variability of chlorophyll in the Arabian Sea through the atmospheric input of nutrients, especially iron. See for example the studies of Wiggert and Murtugudde (2007), Patra et al (2007), Naqvi et al (2010) and Banerjee and Prasanna Kumar (2014). The authors need to at least address the role of dust-induced Fe input in driving the inter-annual variability of chlorophyll in their study region.*

We agree that this should have been discussed in the manuscript and revised the paper accordingly. In the revised model description section, we first clearly acknowledge that the model forcing uses climatological iron inputs (P5 L16-18). We also included a paragraph in the discussion section that acknowledges the potential influence of iron fertilisation by dust storms, quoting the papers listed by the reviewer (P13 L23-34). In this paragraph, we further mention that, despite the fact that non-seasonal iron aerial deposition is not included in our model, it is able to accurately capture the observed interannual variability of the chlorophyll bloom in winter (see Figure 7a), suggesting that this mechanism is unlikely to play a dominant role in the interannual variability of the bloom. We also mention modelling results of Aumont et al. (2008) that suggested that the variability of surface chlorophyll induced by

*The referee comments are italicized.* Answers in regular typeface. Actions taken in red.

the interannual variability of aerosol iron is likely to be very small everywhere, especially relative to the impact of the ocean dynamics because largest fluctuations of surface iron produced by dust occur in oligotrophic regions where phytoplankton growth is not primarily controlled by iron availability. We finally suggest that the mismatch between model and observations for a couple of winters (like in 1997-1998) may be related to the absence of interannual variability of iron deposition, which is not included in the model.

Related References:

Aumont, O., Bopp, L., Schulz, M.: "What does temporal variability in aeolian dust deposition contribute to sea-surface iron and chlorophyll distributions?" Geophysical Research Letters, 35, 7, 2008.

*Minor concerns:*
*3. Page 7 Line 15 " The simulation captures the surface chlorophyll seasonal cycle in the NAS. . ...". While it is so during winter (even in winter note that the model SChl does not capture the increase from November to January seen in the observation), the model completely misses the declining SChl trend from July to August. Instead model depicts the continuation of monotonic increase from May to August. Authors need to point this out while discussing the simulation results.*

We point this out when describing Figure 6 in the revised manuscript (P7 L24-26).

*4. Page 8 lines 7-9 "The figure illustrates that the observed interannual SChl. . ..". The authors need to explain in the text/discussion what the inverse relationship between SChla and MLDa during 2003 & 2005 means.*

Thank you for pointing this out. 2003 and 2005 actually behave inconsistently relative to other years in both the model and observations (Figures 7ab and 8a), suggesting that another mechanism the one that we describe could be at work during these years. We now point it briefly here but have expanded the discussion of other possible mechanisms that could contribute to chlorophyll interannual anomalies in the discussion section, including the potential influence of anomalous iron deposition, as discussed above.

*5. Page 9 lines 2-3 "As a result, the mixed layer nitrate. . ..."Though the MLD nitrate concentration during 2008 Feb is double than that of 2007, the Chlorophyll did not show a commensurate increase. The authors need to explain this in the discussion in the of Redfield ratio and carbon fixation.*

There is no reason for the nitrate variations to be proportional to the chlorophyll variations as biogeochemical models are highly non-linear. In the model, the relation between the phytoplankton and nutrient growth rate is only linear for weak concentrations. In addition, even if the growth rate is increased by a factor two, there is no reason for the chlorophyll

concentration to be doubled. It depends on plenty of other factors, including light availability and grazing rate.

*The referee comments are italicized.* **Answers in regular typeface.** Actions taken in red.
*The study, using multiple datasets and a model, examines the processes related to the interannual variability of winter chlorophyll in the northern Arabian Sea. The Arabian Sea, especially the coastal-open ocean upwelling regions are subjected to an interplay of oceanic processes, such as coastal upwelling, Ekman pumping, mixing, entrainment and lateral/vertical advection (Vialard et al 2012). Other than these physical processes, changes in light penetration and nutrient supply also regulate the chlorophyll content in this basin. Hence it is an interesting task to investigate the biophysical interactions over this region, which result in the observed chlorophyll variations. However, other than presenting several correlation analysis between the mixed layer depth and chlorophyll variations, the study does not delve in depth into the processes governing the chlorophyll changes. The paper is neatly written but it is difficult for the reviewer to comprehend what the prime objectives of the study are, or if there is anything novel in the results. Due to these shortcomings, and as detailed below, I do not recommend the manuscript for publication at its current state.*

As underlined by the first reviewer, our study "*assumes importance because, while the seasonal chlorophyll variations in the Arabian Sea are rather well described, their interannual variability is largely unexplored because of the limitation in long time-series data.*" As we pointed out in the introduction (P3 L1-13), there have indeed been to date only two studies addressing the interannual chlorophyll variations in the northern Arabian Sea in winter and their related mechanisms (Prasanna Kumar et al., 2001 and Wiggert et al., 2002). These studies proposed different mechanisms on the basis of a very limited temporal sampling (two one-month-long in-situ time series in February 1995 and 1997 for Prasanna Kumar et al., 2001) and three consecutive winters from 1998 to 2000 for Wiggert et al., 2002). Our study tests the hypotheses of those two studies based on a much longer dataset (12 winters in observations and 20 winters in the model). In addition, while Wiggert et al. (2002) derived interannual thermocline variations from a model simulation, we derive these variations directly from in-situ (Argo) observations. Both our observational and model results support the Prasanna Kumar et al. (2001) hypothesis rather than the one of Wiggert et al. (2002). It hence better ascertain the mechanisms at stake in driving the chlorophyll interannual variability in the northern Arabian Sea. Another added value of the present study is to provide for this first time an intercomparison of all existing remotely sensed chlorophyll products in this region. We thus believe that our study brings new insights for this highly productive region with no consensus on the mechanisms responsible for year-to-year variations. We underline those points better in the introduction (P3 L12-25) and discussion (P12 L20-27) sections of the revised manuscript.

*The referee comments are italicized.* Answers in regular typeface. Actions taken in red.

Related References:

Prasanna Kumar, S.., Ramaiah, N., Gauns, M., Sarma, V., Muraleedharan, P.M., Raghukumar, S., Kumar, M.D., Madhupratap, M. : Physical forcing of biological productivity in the northern Arabian Sea during the Northeast Monsoon, Deep-Sea Research II 48, 1115–1126, 2001.
Wiggert, J. D., Murtugudde, R., and McClain, C. R.: Processes controlling interannual variations in wintertime (northeast monsoon) primary productivity in the central Arabian Sea, Deep Sea Res., Part II, 47,2319 – 2343, 2002.

*Specific Comments:*
*1. The study revolves around a hypothesis put forward by Wiggert et al (2002) that the interannual variations of chlorophyll intensity is regulated by diurnal mixing.*

As pointed above, we examine two previously proposed mechanisms: that proposed by Wiggert et al. (2002) but also that suggested by Prasanna Kumar et al. (2001).

*The current study, using correlation analyses, points out that the winter chlorophyll variability is tied to the mixed layer depth anomalies, which are associated with the surface heat flux anomalies. However, this factor has already been pointed by studies like Prasanna Kumar et al (2002), where they point out that surface cooling (due to evaporation) in the northern Arabian Sea, combined with reduced incoming solar radiation and high salinity, drives convective mixing, resulting in the upward transport of nutrients in the mixed layer. Similar to the current study, this study also compares two anomalous years and describe the processes involved.*

We think that the current study has several merits:
- There are currently two contradictory hypotheses about the physical control of interannual chlorophyll variability in winter in the Northern Arabian Sea (Wiggert et al., 2002 vs Prasanna Kumar et al., 2001), which we examine separately, finding more support for the latter than for the former.
- Wiggert et al. (2002) and Prasanna Kumar et al. (2001) work are respectively based on 3 and 2 winters. As opposed to the reviewer statement, we do not only analyse two anomalous years but examine long datasets that span 2002-2013 (12 years) derived from in-situ observations for MLD and thermocline and several satellite datasets for chlorophyll along with outputs from a model simulation that spans 1993-2012 (20 years), which considerably strengthens the robustness of our results.
- We also provide an intercomparison of chlorophyll datasets in this region, which was necessary, given the large differences among the datasets. Stating that chlorophyll datasets generally agree for the phase of signals but not their amplitude is an important and useful information for studying chlorophyll variability in the Arabian Sea.

*2. Also, apart from the correlation analyses between mixed layer depth (and nutrients) and chlorophyll concentrations, the manuscript does not examine how the oceanic processes such as the Ekman pumping, offshore advection etc plays a role on the interannual variability.*

*The referee comments are italicized.* Answers in regular typeface. Actions taken in red.

To address the reviewer comment, we performed composite analysis similar to Figure 10 but looking at the wind stress curl (i.e. the driver of Ekman pumping) and surface currents (which could induce advective changes) anomalies associated with enhanced winter blooms. As shown on Figure R2-1 below, this analysis show that interannual chlorophyll fluctuations are not associated with coherent Ekman pumping and surface currents variations in the northern Arabian Sea (while there is a clear MLD signal, see Figure 10b). We also found no significant correlations between interannual anomalies of the wind curl and surface currents with chlorophyll anomalies in the NAS box. These extra-analyses hence suggest that advection of chlorophyll and/or nutrient is unlikely to play a strong role on the interannual fluctuations of the winter bloom in the AS. These analyses are mentioned explicitly in the updated version of the manuscript (P12 L10-16), referring to Figure R2-1 as not shown.

[Figure]

**Figure R2-1:** Observed interannual anomalies of **(a)** ERAI wind stress curl (color) and wind stress (vectors) for composite surface chlorophyll blooms (highlighted on Figure 8). Like for MLD and TCD composites, wind stress curl and wind stress composite are built from the months of February of the same year. **(b)** Same as left, but for surface currents anomalies derived from GEKCO product (Bonjean and Lagerloef, 2002). Regions where composite values are less than the standard error are displayed in white.

Related References:

Bonjean, F., Lagerloef, G.S.E. : Diagnostic Model and Analysis of the Surface Currents in the Tropical Pacific Ocean. J. Physical Oceano., 2, 10, 2938-2954, 2002.

*Also, to counter the study by Wiggert et al (2002), the current study does not examine the contributions due to diurnal mixing.*

We do not explicitly investigate the diurnal cycle in observations because the Argo data are too sparse to be able to analyse the diurnal variability from this dataset. A proper investigation of the impact of diurnal variability on interannual chlorophyll variations from observations would require continuous and long-term temperature and chlorophyll profiles from a fixed location, which are not available to date. Yet, if Wiggert's mechanism was dominating, there should be a negative correlation between interannual variations of chlorophyll and thermocline depth (i.e. a deeper thermocline leading to lower chlorophyll concentrations operating through daily dilution): Table 2, 3 and Figure 8, 9, 10 clearly demonstrates it is not the case in observations, neither over the entire winter period, nor during the beginning (December-January) or the end (February-March) of the winter monsoon period. In addition, despite the absence of a diurnal cycle in the model forcing (i.e. by construction, the model can't reproduce the Wiggert's mechanism), the model shows a good agreement with observed interannual chlorophyll variability in winter in the northern Arabian Sea (see Figures 7a and 10), which is an indirect evidence that the bulk of interannual chlorophyll variations are not linked to the diurnal cycle mechanism. Instead, we find large and statistically significant correlations between MLD and chlorophyll variations at the end of the winter monsoon, suggesting that the mechanism proposed by Prasanna Kumar et al. (2001) (a modulation of nutrient entrainment into the mixed layer, i.e. a mechanism similar to the one proposed at seasonal timescales) dominates during that period. During the early winter monsoon, neither of the two mechanisms appears to drive the weaker interannual chlorophyll anomalies during that season. These points were already mentioned in the submitted version of the paper but this discussion is expanded in the revised manuscript to better stress these points out (P12 L29- P13L21).

*In fact, it is not clear to me where Wiggert et al (2012) says that mixed layer variations are not important in controlling the chlorophyll concentrations – though the study says that diurnal changes of mixed layer are important.*

The mechanism proposed by Wiggert et al. (2002) might indeed imply a correlation between the MLD anomalies and the chlorophyll anomalies. But it also implies a control of MLD and chlorophyll anomalies by the thermocline depth, which is not seen in our results (see L5-6 of the abstract of Wiggert et al., 2002). We point this out better in the discussion section of the revised version (P12 L33- P13 L5).

*3. Page 10, Line 10 says the MLD deepening is controlled by convective overturning, which in turn is controlled by surface heat fluxes. It is not clear to me how the cause and effect is separated here. Surface net heat flux is inversely proportional to the mixed layer depth. Hence it is not surprising as in Fig.10 that they show a good correlation.*

There is a misunderstanding here. We agree with the reviewer that the sea surface temperature (SST) tendency induced by atmospheric heat flux forcing into the ocean (the term Qnet/(rho*Cp*MLD)) could be anticorrelated with MLD variations because the MLD appears in the two terms. However, the y axis of Figure 12c does not show Qnet/(rho*Cp*MLD) but the net heat flux at the ocean-atmosphere interface Qnet and there is no reason for the surface

*The referee comments are italicized.* Answers in regular typeface. Actions taken in red.

net heat flux to be inversely proportional to the mixed layer depth. Indeed surface net heat flux is controlled by atmospheric parameters (wind speed, cloudliness, stability…), which are not controlled by MLD variations. On the other hand the surface heat fluxes exert a strong control on the mixed layer depth. We clarify this in the revised manuscript.

*4. Page 12, Line 20 suggests that the interannual variations in the surface flux are modulated by ENSO (strong correlations). The connection with ENSO was shown by Murtuggude et al (1999), which examined the chlorophyll changes in the Indian Ocean with respect to the 1997-1998 El Niño and the 1998 La Niña. The El Niño – La Niña episodes were accompanied by changes in chlorophyll over the Arabian Sea, with low Chl concentrations during the El Niño period, followed by anomalously high concentrations during the La Niña episode. These changes were attributed to local ocean-atmospheric dynamics linked to the shifts in the Walker circulation. On a similar case, the authors compare 2007 and 2008 MLD and chlorophyll. 2006/7 was a weak El Niño year and 2007/8 was a La Niña year, which is clearly reflected in the Chl anomalies, with the former resulting in negative anomalies and the latter in positive anomalies. Going by correlations as in the current study, I can say that ENSO is a major component in driving both the surface flux and chlorophyll anomalies in the Arabian Sea, on interannual timescales.*

Thanks for pointing this out. We have expanded the discussion about the remote control of ENSO onto the Arabian Sea climate and biogeochemistry in winter in the revised discussion. We will now refer to Murtuggude et al. (1999) who showed that surface chlorophyll was anomalously high during December 1997 and January 1998, consistent with the elevated entrainment flux of nutrients into the euphotic zone expected during stronger than usual northeast monsoons. This indeed suggests from a single event that the 1997/98 El Niño could be responsible for this. We will however also refer to Wiggert et al. (2009) results that show a contrasted biological response of the western Arabian Sea to the 1997/1998 and 2006/2007 El Niño events, with an overall decrease of productivity during the earlier and a slight increase during the latter. Finally, we will also discuss the results from Currie et al. (2013) that showed ENSO control on chlorophyll interannual variations in the Arabian Sea during fall and winter is larger than that of the IOD. In line with these results, our analysis shows that interannual variations of the 2-m air temperature in winter in the northern Arabian Sea box is partly correlated with El Nino-Southern Oscillation (ENSO) index and weakly correlated with Indian Ocean Dipole (IOD) index. The positive correlation sign is however opposite to Murtuggude et al. (1999) results as it indicates that an El Niño event should lead to a reduced winter monsoon. To further explore this aspect, Table R2-1 shows the winter T2m correlation with ENSO index for two different periods: it illustrates that the influence of ENSO on the AS in winter is not very stable in time with large correlations when calculated over the recent 2002-2011 period (0.77) and far weaker correlations when performing this analysis over the extended 1993-2011 period (0.27). This table also shows that all observed and model chlorophyll products do show a negative correlation between ENSO and interannual fluctuations of the winter bloom, consistent with the hypothesis that an El Niño event drive a weaker winter monsoon and hence a weaker bloom. The level of correlation is however modest in all datasets (ranging from -0.11 in AVW to -0.44 in GSM), indicating that ENSO is

not the only driver of the interannual winter bloom variations in this region. These new results along with their consistency with previous literature will be discussed in a new paragraph in the discussion section that details ENSO influence on the Arabian Sea in winter (P14 L17-P15 L8). We also include Table R2-1 as a new Table in the revised manuscript (Table 4).

|  | IOD (SON) | ENSO (NDJ) |
|---|---|---|
| T2a (DJFM 1993-2011) | -0.06 | 0.27 |
| T2a (DJFM 2002-2011) | 0.05 | **0.77** |
| SChla_OC-CCI (2002-2011) | 0.07 | -0.34 |
| SChla_MODIS (2002-2011) | 0.26 | -0.39 |
| SChla_MERIS (2002-2011) | 0.04 | -0.37 |
| SChla_GSM (2002-2011) | 0.24 | -0.44 |
| SChla_AVW (2002-2011) | 0.05 | -0.11 |
| SChla_Model (2002-2011) | 0.02 | -0.30 |

**Table R2-1:** Correlation of IOD and ENSO index with NAS box averaged winter (DJFM) surface temperature anomaly (T2a) and surface chlorophyll anomaly (SChla) derived from different satellite products and model.

Related References:

Currie, J., Lengaigne, M., Vialard, J., Kaplan, D., Aumont, O., Maury, O.: Indian Ocean Dipole and El Niño/Southern Oscillation impacts on regional chlorophyll anomalies in the Indian Ocean. Biogeosciences 10:5841– 5888, 2013.

Murtugudde, R., McCreary, J. P., and Busalacchi, A. J.: Oceanic processes associated with anomalous events in the Indian Ocean with relevance to 1997 –1998, J. Geophys. Res., 105, 3295–3306, 2000.

Wiggert, J. D., Vialard, J., and Behrenfeld, M. J.: Basinwide modification of dynamical and biogeochemical processes by the positive phase of the Indian Ocean Dipole during the SeaWiFS era, in: Indian Ocean Biogeochemical Processes and Ecological Variability, vol. 185, edited by: J. D. Wiggert, R. R. Hood, S. Wajih, A. Naqvi, K. H. Brink, and S. L. Smith, p. 350, 2009.

*5. Also, how do the changes in Eurasian winds (Page 13, Line 8, Goes et al 2005) compare with the ENSO impact? Are the winds increasing, and do they have an impact on the Chl in the Arabian Sea? If so, how does it ride on the interannual variability imposed by ENSO?*

To have a preliminary look into the long-terms trends in winter in the Arabian Sea, Table R2-2 provides the linear trends of wind and chlorophyll in the Arabian Sea. This Table indicate that winds exhibit an increasing trend over the 2002-2011 period. When calculated over an extended period (1993-2011), these winds however show a decreasing trend. Regarding the chlorophyll trends, there is no consistent trend amongst products, some of them showing a decreasing trend (OC-CCI, MERIS, AVW) some others showing an increasing trend (MODIS, GSM). These discrepancies and uncertainties related to the short data length prevent a robust assessment of these trends. We however performed the analyses in the paper using

detrended data and found that our results regarding the interannual variability are robust. This is illustrated on Figure R2-2 that shows an analysis similar to that of Figure 9 but for detrended data. This is pointed out in the revised version (P15 L8-12).

| Dataset | Period | Trend |
|---|---|---|
| Wind-ERAI | 1993-2011 | -0.006 |
| Wind-ERAI | 2002-2011 | 0.04 |
| SChla_OC-CCI | 2002-2011 | -0.005 |
| SChla_MODIS | 2002-2011 | 0.015 |
| SChla_MERIS | 2002-2011 | -0.007 |
| SChla_AVW | 2002-2011 | -0.042 |
| SChla_GSM | 2002-2011 | 0.03 |
| SChla_Model | 1993-2011 | -0.0015 |
| SChla_Model | 2002-2011 | 0.0023 |

**Table R2-2:** Linear trends for NAS box averaged winter (DJFM) surface wind speed anomaly from ERA-Interim and surface chlorophyll anomaly (SChla) derived from different satellite products and model.

[Figure]

**Figure R2-2:** Scatterplot of winter (DJFM), NAS-averaged detrended OC-CCI surface chlorophyll anomalies against observed detrended **(a)** MLD and **(b)** thermocline depth anomalies. **(c-d)** Idem for Model.

**Physical control of the northern Arabian Sea winter chlorophyll bloom interannual variations**

M.G. Keerthi[1], Matthieu Lengaigne[2, 3], Marina Levy[2], Jerome Vialard[2], V. Parvathi[1], Clément de Boyer Montégut[4], Christian Ethé[2], Olivier Aumont[2], I. Suresh[1], V.P. Akhil[1], P.M. Muraleedharan[1]

[1] CSIR-National Institute of Oceanography (CSIR-NIO), Goa, India

[2] Sorbonne Universités (UPMC, Univ Paris 06)-CNRS-IRD-MNHN, LOCEAN Laboratory, IPSL, Paris, France

[3] Indo-French Cell for Water Sciences, IISc-NIO-IITM–IRD Joint International Laboratory, NIO, Goa, India

[4] IFREMER, Univ. Brest, CNRS, IRD, Laboratoire d'Océanographie Physique et Spatiale, IUEM, 29280, Brest, France

*Correspondence to*: M. G. Keerthi (keerthanaamg@gmail.com)

**Abstract.** The northern Arabian Sea hosts a winter chlorophyll bloom, triggered by convective overturning in response to cold and dry northeasterly monsoon winds. Previous studies of interannual variations of this bloom only relied on a couple of years of data and reached no consensus on the associated processes. The current study aims at identifying these processes using both ~10 years of observations (including remotely-sensed chlorophyll data and physical parameters derived from Argo data) and a 20-year long coupled biophysical ocean model simulation. Despite discrepancies in the estimated bloom amplitude, the six different remotely-sensed chlorophyll products analysed in this study display a good phase agreement at seasonal and interannual timescales. The model and observations both indicate that the winter bloom interannual fluctuations are strongly tied to mixed layer depth interannual anomalies (~0.6 to 0.7 correlation), which are themselves controlled by the net heat flux at the air-sea interface. Our modelling results suggest that the mixed layer depth control of the bloom amplitude ensues from the modulation of nutrient entrainment into the euphotic layer. In contrast, the model and observations both display insignificant correlations between the bloom amplitude and thermocline depth, which precludes a control of the bloom amplitude by daily dilution down to the thermocline depth, as suggested in a previous study.

**1. Introduction**

Located in the western arm of the northern Indian Ocean, the Arabian Sea (AS) is forced by energetic seasonally reversing monsoon winds, which largely control its physical properties. During boreal summer, strong southwesterly winds blow over the western AS (Findlater, 1969) and cause intense upwelling along the coasts of Somalia and Oman and downwelling in the central AS (e.g. Schott and McCreary, 2001). During boreal winter, the Eurasian continent cools and a high-pressure region develops on the Tibetan plateau, resulting in cold and dry north/northeasterly winds (e.g. Smith and Madhupratap, 2005) and leading to strong evaporative cooling (Dickey et al., 1998). These diverse physical processes cause substantial variations in marine biogeochemical and ecosystem response. Being one of the most productive regions in the world ocean (Satya Prakash and Ramesh, 2007; Prasanna Kumar et al., 2000) and being home to the second most intense oxygen minimum zone in the world ocean (Kamykowski and Zentara, 1990), the AS provides an excellent test bed for studying biophysical coupled processes (McCreary et al., 2009).

Previous studies have extensively described the seasonal variability of surface chlorophyll in the AS. The AS biogeochemical properties vary from stratified oligotrophic conditions during inter-monsoon periods to eutrophic conditions during monsoons (Smith et al., 1998; McCreary et al., 2009). Neither surface irradiance nor temperature limits the biological productivity in this tropical basin: instead, it is mostly attributed to dynamical processes in response to the monsoonal forcing (e.g. Barber et al., 2001; Marra and Barber, 2005). During boreal summer, the largest seasonal blooms are found along the coasts of the Arabian peninsula (e.g. Banzon et al., 2004; Lévy et al., 2007; Wiggert et al., 2005) and are exported offshore by mesoscale eddy stirring (e.g. Resplandy et al., 2011). In boreal winter, convective overturning allows entrainment of nutrients into the mixed layer and leads to a prominent bloom in the northern AS (Banse and English, 2000; Madhupratab et al., 1996; Prasanna Kumar et al., 2001; Wiggert et al., 2002). In addition to these seasonal variations, several studies revealed large interannual variations in the AS winter chlorophyll from either satellite (Banse and McClain, 1986; Banse and English, 1993; Sarma et al., 2006; Wiggert et al., 2002; Sarma et al., 2012) or in-situ measurements (Bauer et al., 1991; Madhupratap et al., 1996; Gundersen et al., 1998; Prasanna Kumar et al., 2001). This strong interannual variability of the northern AS winter bloom is illustrated on Fig.1a,b for two consecutive winters. A particularly intense bloom developed in the northern AS during winter 2007 (Fig. 1b), with high surface chlorophyll concentration (>1.0 mg.m$^{-3}$) extending southward down to 14°N. In contrast, the winter 2006 bloom remained confined to the northern AS (Fig. 1a), with high chlorophyll concentration (>1.0 mg.m$^{-3}$) limited to the north of 20°N. The difference in the amplitude of the bloom between winter 2006 and 2007 averaged over the northern AS box (hereafter NAS region shown in Fig. 1) reaches 0.22 mg.m$^{-3}$, which is approximately 30% of the climatological winter chlorophyll value.

Understanding the mechanisms driving these chlorophyll interannual variations is important, as this might have a profound influence on the variations of the fish stocks and of the oxygen minimum zone in the AS. To date, only a few

keerthimg 3/8/y 4:10 PM

keerthimg 3/8/y 4:10 PM

keerthimg 3/8/y 4:10 PM

keerthimg 3/8/y 4:10 PM

keerthimg 3/8/y 4:10 PM

keerthimg 3/8/y 4:10 PM

[revised manuscript text omitted]

---

## Referee Report (RR1)

Manuscript Number: 10.5194/bg-2016-153, 2016, Revised version.

Title: Physical control of the northern Arabian Sea winter chlorophyll bloom interannual variations

Authors: Keerthi et al.

**Reviewer's opinion:**

The reviewer appreciates the careful revision by the authors, and the extended analysis provided for substantiating their results. I agree that it is important to clarify the mechanisms governing the interannual variability of winter chlorophyll over the productive Arabian Sea region. The manuscript may be considered for publication, but only after some revisions and clarifications.

1. Page 3, Line 8. The manuscript says that "Wiggert et al. (2002) suggested using a simple one-dimensional model, that interannual variations of the bloom intensity were not controlled by interannual MLD variations". Again, I am not sure if Wiggert et al. implied that mixed layer depth variations are not important in controlling the chlorophyll interannual variations (please show me which part of their manuscript implies that). Wiggert et al. study, from my understanding, indicates that diurnal variability of mixed layer is important. It seems to me that the current results are complimenting, and not contradicting Wiggert et al. or Prasanna Kumar et al. (2001)

2. Page 15, Line 18. This section requires revision. The current study is built on the logic that the earlier studies used only a few years of data, which is insufficient to understand the interannual variability of chlorophyll. Meanwhile, the study confidently refers to Goes et al. (2005), which talks about trends in chlorophyll and winds using merely 7 years of data, which is inadequate to extract trends without endpoint sensitivity (Beaulieu et al. 2013). Also, out of these seven years, the first year was an El Niño year, which played a major role in the reduction of chlorophyll concentrations in the Arabian Sea (Murtugudde et al. 1999), and skewing the time series. Besides, several studies (e.g. Roxy et al. 2015) indicate that the monsoon winds are exhibiting a weakening trend, though the changes over the western Arabian Sea are insignificant or uncertain. Using long-term observations and model simulations, a recent study (Roxy et al. 2016) does indicate that the summer chlorophyll concentrations are reducing as a result of stratification due to surface warming. Hence the authors need to update the discussion based on these recent studies.

References:
Beaulieu, C., S. A. Henson, J. L. Sarmiento, J. P. Dunne, S. C. Doney, R. Rykaczewski, and L. Bopp (2013), Factors challenging our ability to detect long-term trends in ocean chlorophyll, Biogeosciences, 10, 2711–2724.
Murtugudde, R. G., Signorini, S. R., Christian, J. R., Busalacchi, A. J., McClain, C. R., & Picaut, J. (1999). Ocean color variability of the tropical Indo-Pacific basin observed by SeaWiFS during 1997–1998. Journal of Geophysical Research: Oceans, 104(C8), 18351-18366.

Roxy M. K., A. Modi, R. Murutugudde, V. Valsala, S. Panickal, S. Prasanna Kumar, M. Ravichandran, M. Vichi and M. Levy, 2016: A reduction in marine primary productivity driven by rapid warming over the tropical Indian Ocean. Geophys. Res. Lett., 43, 826-833.

---

## Author Response (AR2)

*The editor comments are italicized.* Answers in regular typeface. Actions taken in red.

Manuscript number: 10.5194/bg-2016-153, 2016, Revised version.

Title: Physical control of the northern Arabian Sea winter chlorophyll bloom interannual variations

Authors: Keerthi et al.

Editor Comments

*I have read the revised version of your manuscript as well as the comments of the two reviewers on this revised version and on the answers that you have provided to their criticisms. Reviewer 1 is not satisfied by your answers to his/her criticisms while reviewer 2 suggests some new improvements and does not agree with your conclusions that your work contradicts Wiggert et al (2002) findings.*

We thank the editor very much for his careful reading of the manuscript and reviewers comments. We are surprised by the second round of comments from reviewer 1. Reviewer 1 did positively evaluate our manuscript in his first round of review, with only two major concerns (winter season definition; potential role of dust storms). We did change all the analyses, figures and related text to use a more appropriate season definition based on his comment and added an extended discussion on the potential role of dust storms (which does not vary interannually in our experiments). In the second round, the same reviewer was very succinct and stated that we did not address his major comments, without giving any further details. He also raised some concerns about the originality of our approach, which he did not previously mention. Due to the lack of specific suggestions, we are not in a position to answer his second round of comments. We will hence focus on addressing your concerns in the point-by-point answer below. We have of course also addressed the comments of reviewer 2 (see separate point-by-point answer), which was rather critical during the first round of review but evaluated our revised manuscript positively.

*Based on that I recommend major revisions of your manuscript and, in particular I would like you to tackle the following points:*

*In his/her first review, reviewer 1 would have liked you to address the role of dust-induced Fe input in driving the interannual variability of chlorophyll. In some part of your answers you say*

*The editor comments are italicized.* Answers in regular typeface. Actions taken in red.

*that it is not important for driving interannual variability while in other places you suggest that finally it may be important (2002 and 2004). In the revised version, you give arguments justifying your approach based on Aumont et al., 2008 results. I would suggest that 1) in the description of the model you justify why you impose a climatological seasonal cycle without interannual variability for dust storm (e.g. as mentioned in the title the scope of your paper is to address the physical control, and this choice was made supported by Aumont et al., 2008 findings); 2) in the discussion, I would like that you envisage to look whether the year 2002 and 2004 which do not seem to obey to the physical control have particular dust storm events in your region (I do not like the last sentence of page 13, which is vague, see my comment below); this could also be helpful that in the introduction when you enumerate the different factors that may control the interannual variability of chlorophyll in the region you mentioned that dust storm (+references) and justify why you decide to focus on the physics (reference to Aumont et al 2008).*

Following your suggestions, the use of a climatological forcing for dust deposition is now better justified in the model description section (P5 L27-33). We further highlighted that no dust storms were reported for the winters of 2002 and 2004 in the discussion section (P14 L33 - P15 L7), which suggests that this specific mechanism may not be responsible for the peculiar SChl behaviour during those years. In addition, we further highlight that our work and that of Banerjee and Prasanna Kumar (2014) focus on two different regions, namely the northern AS where winter convection plays a strong role and the central AS where no convective overturning occurs. The different physics of the two regions may imply a different role of iron limitation, potentially explaining the mismatch between our study (for which Fe interannual variations don't seem to play a big role) and that of Banerjee and Prasanna Kumar (2014).

*Your answers to reviewer 1 comments concerning the differences of 2007 and 2008 can also be considered as vague, even if the relationship is indeed non-linear some explanation on why the chlorophyll does not follow the increase of nitrate can be found with your modeling tools (this is one of the aims of models). This will allow to further highlight the added value compared to past studies in terms of the understanding of mechanisms.*

To clarify the relationship, we have modified Figure 11 to include curves of surface chlorophyll and surface and subsurface nitrate concentrations. This now clarifies that there is no nonlinearity in the modeled chlorophyll response to nitrate, when considering the full length of the bloom

*The editor comments are italicized.* Answers in regular typeface. Actions taken in red.

(which ends in February in 2007 and in March in 2008). The two times larger February 2008 nitrate concentration maintains new chlorophyll production until March 2008. In other words, it takes more time for phytoplankton to exhaust nitrate in the mixed layer in 2008. The paragraph corresponding to Figure 11 now explains this better (P10 L23- P11 L5).

*Reviewer 2 does not agree that conclusions of your work contradict those of Wiggert et al., 2002. There is a disagreement between the reviewer's interpretation and yours. This is important to clarify that point since in a lot of places in the manuscript (already in the abstract) you mentioned that your findings contradict those of previous studies. In particular, I would like that you comment on the possible limitations of correlating averaged (in space and time) values of MLD, TCD and Schl, that you more thoroughly justify your estimations of TCD (and why you consider that TCD it is a good indicator of the daily variations of the MLD), comment on the capacity of the data that you have to assess TCD at this time scale.*

Wiggert et al. (2002) clearly point out that the mechanism they propose should lead to a negative correlation between TCD and SChl anomalies, i.e. an unusually deep winter TCD leads to a weak bloom amplitude through daily dilution. In their discussion section (page 2340), Wiggert et al. (2002) state that, during the winter monsoon, « *primary productivity is never nutrient limited and a deeper thermocline will result in lower values of mixed layer Chl a, due to the deeper penetration of the diurnal mixed layer* ». They further acknowledge at page 2338 that « *Such an inverse relation* (between SChl and thermocline depth) *runs counter to the standard paradigm of deeper mixing enhancing nutrient concentration that in turn leads to increased primary production* ». Wiggert et al. (2005) further indicate at page 197-198 that « *Wiggert et al. (2002) demonstrated that higher chlorophyll concentration coincided with a shallower thermocline, a relationship that directly contradicts the Bermuda paradigm. During the NEM, a deeper thermocline allows deeper diurnal mixing, which results in greater daily dilution of euphotic zone phytoplankton biomass and stronger inhibition of bloom development (see Fig. 9 in Wiggert et al., 2002). Kumar et al. (2001b) also investigated interannual chlorophyll variability (NEMs of 1995 and 1997) and concluded that the Bermuda paradigm was operating* ». This clearly underlines that our main results, which are in line with Prasanna Kumar et al. (2001), contradict those of Wiggert et al. (2002). To further clarify the relationship between SChl, MLD and TCD variability implied by these two studies, we expanded the introductory section (P3 L2-7 and P3

*The editor comments are italicized.* Answers in regular typeface. Actions taken in red.

L9-17). We also expanded the method section, by further justifying why we investigate MLD and TCD variations, and the expected relationship with SChl for each of the two proposed mechanisms (P4 L17-22). Regarding possible limitations of correlating average values, we added a remark that the link between SChl/MLD (and absence of link between SChl/TCD) remains valid over the entire NAS box when describing the composite of figure 10 (P10, L12-13). Finally, as suggested by the editor, we added a paragraph in the discussion section, addressing the potential limitations and uncertainties related to our observational analysis (P13 L19-29).

*Some other comments:*

*Introduction line 14, Schl needs to be defined there because you use it afterwards in Figure 1 before its definition in section 2.*
Done

*Figure 6: please explain how these averaged curves have been produced for the model MLD, TCD (model results averaged at observation points? Or averaged over the whole domain?). If I am right, you are not modeling the wind (this is confusing to have it in the model as well) what about the nitracline depth? Is it from the data? If yes, again how the model average is computed? What about the heat flux (mode vs data)?*
Nitracline depth is an output from our model (there is currently no available observational interannual nitracline depth gridded dataset). The figure 6 model curves are averaged over the entire NAS region. The Figure 6 (and 7) captions have been edited accordingly. As suggested by the editor, the wind and heat flux are indeed input variables of the model (and not model outputs). We hence removed panel 6f to avoid confusion.

*Section 4: "The hypotheses of Wiggert et al. (2002) and Prasanna Kumar et al. (2001) for the mechanisms that controls SChl interannual variations imply a correlation of SChl anomalies with TCD and MLD anomalies, respectively". This is where the reviewer does not agree with you. Please make your justification clearer as why a correlation between SChl and diurnal variations of MLD implies a correlation with monthly basin averaged TCD anomalies. This is where there is a disagreement.*

*The editor comments are italicized.* Answers in regular typeface. Actions taken in red.

Cf our answer to your major point. We did expand the paragraph discussing these mechanisms in the introductory section, justifying why Wiggert et al. (2002) mechanism implies a correlation of SChl and TCD interannual anomalies (P3 L9-17)

*Figures 8 and 9: please clarify whether you average the SChl at only points (in space and time) where and when you have observations of MLD and TCD?*

The model curves are produced from an average over the entire box. This is now clarified in the caption.

*Page 13, line 30: "Even though the non-seasonal iron aerial deposition is not included in our model, it is able to ..". The first part of this sentence is not clear, do you mean that the absence of consideration of an interannual cycle? Please reformulate.*

Reformulated (P15 L1-3).

*Page 13, last sentence, "However, the mismatch between model and observations for a couple of winters may be related to the absence of interannual variability of iron deposition, which is not included in the model". I find this sentence in contradiction with what is mentioned above: "Even though the non-seasonal iron aerial deposition is not included in our model, it is able to accurately capture the observed interannual variability of the chlorophyll bloom in winter (Fig. 7a), suggesting that this mechanism is unlikely to play a dominant role in ..". First you say that the model is very good then you say that there are mismatches. Do you mean that the model is usually good expect during some years which can be due to the absence of a consideration of the interannual (variability of iron deposition (please clarify).*

This section was indeed confusing: it has been reformulated (P14 L29- P15 L4).

*Page 14, first paragraph, I do not really capture the main message that the authors want to communicate with this paragraph. Do you mean that one of the main messages of this manuscript (positive correlation between MLD and Schl) depends on model resolution? I like the idea to confront different findings but here the reader is expecting a deeper analysis (e.g. impact of model resolution on model performance and, in particular simulation of MLD, Schl, TCD, which model finding is the "best" one) of these different findings which could be the topic of another*

*paper. Is there a publication that analyses the interannual variability of the MLD and Schl correlation in the NAS based on the global model? If not I find difficult to analyze the different findings in 15 lines, this is confusing.*

We wanted to illustrate that a mean state bias can induce a wrong mechanism in the model. But we agree that this section was not very useful: we have removed it.

*Page 14, lines 27: "Wiggert et al. (2009) results also point to a contrasted biological signature of the western AS during the 1997/1998 and 2006/2007 El Niño events, with an overall decrease of productivity during the earlier and a slight increase during the latter". This sentence seems in contradiction with the one above "Murtuggude et al. (1999) showed that SChl was anomalously high during December 1997 and January 1998, consistent ... » . Please clarify. Do you mean that this is not clear what is the impact of El Nino events on the Schl?*

Murtugudde et al. (1999) in fact pointed to a "high productivity" in the northeastern Arabian Sea, but their plot included the seasonal cycle, and are not relevant to discuss interannual variability.

We have hence removed the reference to Murtugudde et al. (1999) and tightened the rest of the paragraph (P 15, L15-29).

*The referee comments are italicized.* Answers in regular typeface. Actions taken in red.

Manuscript number: 10.5194/bg-2016-153, 2016, Revised version.

Title: Physical control of the northern Arabian Sea winter chlorophyll bloom interannual variations

Authors: Keerthi et al.

*Anonymous Referee #1*

*Reviewer's opinion:*

*I am sorry to state that I am not convinced by the author's rebuttal. In spite of making clear my major concerns explicitly and vividly, the authors did not make adequate efforts to address them squarely. Interestingly, the authors came up with additional figures, tables and also added more elaborate text, but failed to address the basic concerns. In short, the authors did not successfully demonstrate what drives the inter-annual variability of winter chlorophyll bloom. Neither did they come out strongly on how their study is significantly different from previous studies.*

We are surprised by the second round of comments from reviewer 1. Reviewer 1 did positively evaluate our manuscript in his first round of review, with only two major concerns (winter season definition; potential role of dust storms). We did change all the analyses, figures and related text to use a more appropriate season definition based on his comment and added an extended discussion on the potential role of dust storms (which does not vary interannually in our experiments). In the second round, the reviewer was very succinct and stated that we did not address his major comments, without giving any further details. He also raised some concerns about the originality of our approach, which he did not previously mention. Due to the lack of specific suggestions, we are not in a position to answer his second round of comments.

We have hence focused on addressing the specific suggestions by the editor & reviewer 2 (see separate point-by-point answers).

*The referee comments are italicized.* Answers in regular typeface. Actions taken in red.

Manuscript number: 10.5194/bg-2016-153, 2016, Revised version.

Title: Physical control of the northern Arabian Sea winter chlorophyll bloom interannual variations Authors: Keerthi et al.

*Anonymous Referee #2*

*Reviewer's opinion:*

*The reviewer appreciates the careful revision by the authors, and the extended analysis provided for substantiating their results. I agree that it is important to clarify the mechanisms governing the interannual variability of winter chlorophyll over the productive Arabian Sea region. The manuscript may be considered for publication, but only after some revisions and clarifications.*

We thank the reviewer for his positive comments and for his inputs on the paper. We provide a detailed answer to each of the comments below.

*1. Page 3, Line 8. The manuscript says that "Wiggert et al. (2002) suggested using a simple one-dimensional model, that interannual variations of the bloom intensity were not controlled by interannual MLD variations". Again, I am not sure if Wiggert et al. implied that mixed layer depth variations are not important in controlling the chlorophyll interannual variations (please show me which part of their manuscript implies that). Wiggert et al. study, from my understanding, indicates that diurnal variability of mixed layer is important. It seems to me that the current results are complimenting, and not contradicting Wiggert et al. or Prasanna Kumar et al. (2001)*

We agree with the reviewer that Wiggert el. (2002) mechanism does not imply anything about correlations between MLD and SChl interannual variations. This part of the sentence has been removed. However, Wiggert et al. (2002) clearly point out that the mechanism they propose should lead to a negative correlation between TCD and SChl anomalies, i.e. an unusually deep winter TCD leads to a weak bloom amplitude through daily dilution. In the discussion section (page 2340), Wiggert et al. (2002) state that, during the winter monsoon, « *primary productivity is never nutrient limited and a deeper thermocline will result in lower values of mixed layer Chl a, due to the deeper penetration of the diurnal mixed layer* ». They further acknowledge at page

2338 that « *Such an inverse relation* (between SChl and thermocline depth) *runs counter to the standard paradigm of deeper mixing enhancing nutrient concentration that in turn leads to increased primary production* ». Wiggert et al. (2005) further indicate at page 197-198 that « *Wiggert et al. (2002) demonstrated that higher chlorophyll concentration coincided with a shallower thermocline, a relationship that directly contradicts the Bermuda paradigm. During the NEM, a deeper thermocline allows deeper diurnal mixing, which results in greater daily dilution of euphotic zone phytoplankton biomass and stronger inhibition of bloom development (see Fig. 9 in Wiggert et al., 2002). Kumar et al. (2001b) also investigated interannual chlorophyll variability (NEMs of 1995 and 1997) and concluded that the Bermuda paradigm was operating* ». This clearly underlines that our main results, which are in line with Prasanna Kumar et al. (2001), contradicts those of Wiggert et al. (2002). To further clarify the implication of these two studies regarding the relationship between SChl, MLD and TCD variability, we expanded the introductory section (P3 L2-7 and P3 L9-17). We also expanded the method section, by further justifying why we investigate MLD and TCD variations, and the expected relationship with SChl for each of the two proposed mechanisms (P4 L17-22).

*2. Page 15, Line 18. This section requires revision. The current study is built on the logic that the earlier studies used only a few years of data, which is insufficient to understand the interannual variability of chlorophyll. Meanwhile, the study confidently refers to Goes et al. (2005), which talks about trends in chlorophyll and winds using merely 7 years of data, which is inadequate to extract trends without endpoint sensitivity (Beaulieu et al. 2013). Also, out of these seven years, the first year was an El Niño year, which played a major role in the reduction of chlorophyll concentrations in the Arabian Sea (Murtugudde et al. 1999), and skewing the time series. Besides, several studies (e.g. Roxy et al. 2015) indicate that the monsoon winds are exhibiting a weakening trend, though the changes over the western Arabian Sea are insignificant or uncertain. Using long-term observations and model simulations, a recent study (Roxy et al. 2016) does indicate that the summer chlorophyll concentrations are reducing as a result of stratification due to surface warming. Hence the authors need to update the discussion based on these recent studies.*

The discussion on the AS trend has been updated and now includes the suggested references (P

*The referee comments are italicized.* **Answers in regular typeface.** Actions taken in red.

P16 L7-11). Thank you.

References: Beaulieu, C., S. A. Henson, J. L. Sarmiento, J. P. Dunne, S. C. Doney, R. Rykaczewski, and L. Bopp (2013), Factors challenging our ability to detect long-term trends in ocean chlorophyll, Biogeosciences, 10, 2711– 2724.

Murtugudde, R. G., Signorini, S. R., Christian, J. R., Busalacchi, A. J., McClain, C. R., & Picaut, J. (1999). Ocean color variability of the tropical Indo   Pacific basin observed by SeaWiFS during 1997–1998. Journal of Geophysical Research: Oceans, 104(C8), 18351-18366.

[revised manuscript text omitted]

keerthimg 5/26/y 6:57 PM

keerthimg 5/26/y 6:57 PM

keerthimg 5/26/y 6:57 PM

keerthimg 5/26/y 6:57 PM

keerthimg 5/26/y 6:57 PM

keerthimg 5/26/y 6:57 PM

keerthimg 5/26/y 6:57 PM

**2. Data and Method**

**2.1. Observations**

The SChl estimates analysed in the present study are derived from different instruments (SeaWiFS, MERIS and MODIS), retrieval algorithms and span different periods (Table 1). We will compare these different retrievals in Section 3.1, in order to assess the robustness of remotely-sensed data to investigate the NAS winter bloom. We used the Level-3 Standard Mapped Images with a 9x9-km spatial and a monthly temporal resolution downloaded from http://oceancolor.gsfc.nasa.gov

10 for all of these single-mission products. In addition, we also used three level 3 merged ocean-color products downloaded from http://www.oceancolour.org/ at 4x4-km and monthly resolution: the weighted average empirical (AVW) product, the semi-analytical Garver Siegel Maritorena (GSM) product and the Ocean Color Climate Change Initiative (OC-CCI) product. The longest observational period is provided by the OC-CCI product and spans from October 1997 to December 2013.

15 The ocean physical parameters are derived from an updated version of the dataset described in Keerthi et al. (2013), with an extended temporal coverage (2002 to 2013) and an estimate of the TCD in addition to that of the MLD. These MLD and TCD datasets are built from a combination of Argo and historical temperature and salinity profiles. To assess whether the mechanism proposed by Prasanna Kumar et al. (2001; i.e. the Bermuda paradigm) dominates the interannual variability of the winter bloom in the northern AS over this extended period, we will investigate if there is a correlation between in-situ-

20 derived interannual MLD and the satellite-derived interannual SChl anomalies. We will test the alternative mechanism proposed by Wiggert et al. (2002) by investigating if there is a negative correlation between interannual in-situ-derived TCD anomalies and interannual satellite-derived SChl anomalies in the northern AS during winter. MLDs were estimated using a temperature criterion, and are defined as the depth where the temperature increases by 0.2°C with respect to the temperature at 10 m. The reference depth was taken at 10 m to avoid aliasing by the diurnal cycle. The TCD was defined as the depth of

25 the maximal vertical temperature gradient. MLDs and TCDs were estimated from individual temperature profiles at their native vertical resolution. The resolution of the data was then degraded to a regular 2° monthly grid, by taking the median of all MLDs and TCDs in each grid mesh. A more detailed description of this procedure can be found in Keerthi et al. (2013). An overview of the spatio-temporal coverage of this dataset over the NAS is provided in Fig. 2. While the data coverage is particularly sparse in winter before 2002 in our targeted region (e.g. less than 10 data per month are available in winter 2000

30 in the NAS region), the data density increased considerably after 2002, with the development of the Argo program (Fig. 2a). After 2002, the NAS box winter data density ranged from 25 profiles per month during 2005 to nearly 120 profiles per month during 2012. This implies that the interannual MLD/TCD values averaged over the NAS box during winter 2002 to 2013 are built from an average of 100 to 500 individual values, giving us confidence in the robustness of the interannual MLD/TCD variability derived from this in-situ dataset. It should be noticed that the data coverage is however not spatially

keerthimg 5/26/y 6:57 PM

keerthimg 5/26/y 6:57 PM

keerthimg 5/26/y 6:57 PM

keerthimg 5/26/y 6:57 PM

keerthimg 5/26/y 6:57 PM

keerthimg 5/26/y 6:57 PM

keerthimg 5/26/y 6:57 PM

keerthimg 5/26/y 6:57 PM

keerthimg 5/26/y 6:57 PM

[revised manuscript text omitted]

keerthimg 5/26/y 6:57 PM